# CORE-MTL: Rethinking Gradient Balancing
# via Causal Orthogonal Representations

**Chengfeng Wu** [1] **Tao Zou** [2] **Yanru Wu** [1] **Jingge Wang** [1]

## Abstract

Multi-task learning (MTL) aims to construct a joint model for multiple tasks by sharing a common representation across domains. To achieve this goal, existing optimization-centric methods either balance task gradients or modify the shared architecture. However, as these approaches remain agnostic to the content of the shared representation, they fail to disentangle task-relevant structure from spurious context, leading to negative transfer and poor generalization. To overcome this limitation, we propose **C**ausal **O**rthogonal **R**epresentations for **M**ulti-**T**ask **L**earning (CORE-MTL), a causally motivated representation-centric framework that encourages a structured semantic-residual factorization of the shared representation, concentrating task-relevant structure in the semantic stream while relegating nuisance variation to the residual stream. We instantiate this framework in the visual domain by leveraging physical priors for structured scenes and statistical constraints for attributes. Theoretically, our method enjoys a tighter out-of-distribution generalization bound than optimization-centric methods and reduces task gradient interference without explicit gradient projection or reweighting. Empirically, CORE-MTL consistently outperforms existing methods on visual multi-task benchmarks in both in-distribution and out-of-distribution settings. Code is publicly available at https://github.com/Hope-Rita/CORE-MTL.

## 1. Introduction

Multi-task learning (MTL) aims to improve sample efficiency and robustness by training a shared encoder jointly on several related tasks. However, MTL often suffers from negative transfer: optimizing for one task can harm others, and performance can degrade sharply when the data distribution or environment changes. Most existing approaches either try to reconcile task gradients by reweighting or projecting them at each update (e.g., GradNorm (Chen et al., 2018), PCGrad (Yu et al., 2020), STCH (Lin et al., 2024)), or modify the shared architecture so that each task attends to its own slice of the backbone features (e.g., MTAN (Liu et al., 2019)). While both directions have driven progress, we argue that a crucial aspect remains largely overlooked: the internal structure of the shared representation. Most approaches treat feature learning as a black box, ignoring how the entanglement between invariant semantic factors and spurious style factors limits both in-distribution (ID) stability and out-of-distribution (OOD) robustness.

We advocate a representation-level perspective on MTL that focuses on how the shared feature space is structured, rather than on gradient rebalancing or task-specific heads. A key observation is that gradient conflict is often a symptom rather than the root cause of negative transfer (Shi et al., 2023). In practice, tasks are frequently coupled through shared nuisance-driven shortcuts—features that correlate with labels in-distribution but are not causally stable across tasks. When such nuisance variation is encoded in the shared representation, adjusting gradients alone cannot resolve the resulting conflicts. From this perspective, both gradient-balancing and architecture-based methods operate purely at the optimization level—reweighting losses, projecting updates, or changing task routing—without shaping what the shared representation itself is allowed to encode (Hu et al., 2022). As a result, nuisance factors can persist as shared shortcuts, leading to negative transfer and brittle OOD generalization.

Motivated by this perspective, we propose **C**ausal **O**rthogonal **R**epresentations for **M**ulti-**T**ask **L**earning (*CORE-MTL*), a representation-centric framework instantiated for visual multi-task learning that targets a causally motivated semantic–residual factorization that reduces

[1]Shenzhen Key Laboratory of Ubiquitous Data Enabling, Tsinghua Shenzhen International Graduate School, Tsinghua University, Shenzhen, China [2]State Key Laboratory of Complex & Critical Software Environment, Beihang University, Beijing, China. Correspondence to: Tao Zou <zoutao@buaa.edu.cn>.

*Proceedings of the 43rd International Conference on Machine Learning*, Seoul, South Korea. PMLR 306, 2026. Copyright 2026 by the author(s).

nuisance-driven shortcuts under an assumed latent SCM. The method combines a structural decomposition of the shared representation with a training scheme that encourages this decomposition according to the assumed semantic–nuisance factorization. First, on the architectural side, CORE-MTL decomposes the shared features into a *semantic stream* and a *residual stream*. The semantic stream captures task-relevant information that is stable across tasks, while the residual stream absorbs task-irrelevant nuisance variation. Task predictors are built solely on the semantic stream, which prevents nuisance signals from acting as shortcuts during learning. Second, on the optimization side, CORE-MTL employs a unified objective that combines reconstruction, disentanglement, and counterfactual invariance losses to align the decomposition with the causal motivation. To assign clear roles to the semantic and residual streams, we ground the latent factors using physical priors for structured scenes and statistical and architectural constraints for attributes. Hard grounding provides stronger semantic anchoring, while soft grounding supports functional disentanglement. This training scheme encourages nuisance variation to be confined to the residual stream and promotes stable shared semantics, resolving negative transfer at the representation level while remaining compatible with standard MTL pipelines.

Our main contributions are summarized as follows:

- From a causal perspective, we formalize the limitations of optimization-centric methods under an assumed latent SCM, proving that manipulating gradients on entangled representations can lead to an unavoidable OOD error floor. In contrast, our disentangled architecture guarantees a tighter OOD generalization bound and induces gradient orthogonality. This mechanism resolves conflicts at the representation level, allowing distinct tasks to synergize naturally without the need for rigid post-hoc gradient correction.

- We propose *CORE-MTL*, a representation-centric framework instantiated in the visual domain that encourages a structured factorization between task-relevant semantics and nuisance style factors via physical or generic grounding. By aligning task heads to the semantic stream via disentanglement and counterfactual objectives, our method resolves conflicts at the representation level rather than through post-hoc gradient surgery.

- Experiments on three in-distribution and two out-of-distribution visual multi-task benchmarks, comparing *CORE-MTL* to ten representative multi-task baselines, show consistent improvements in both ID and OOD performance while remaining effective as the number of tasks and attributes increases.

## 2. A Structural View of Multi-Task Learning

In this section, we study multi-task learning from a structural perspective, focusing on optimization-centric methods that coordinate tasks only through their gradients. We analyze these methods under a simple causal view of shared representations and distribution shift.

### 2.1. Multi-Task Learning Setup

We consider a multi-task learning problem with $K \geq 2$ tasks sharing an input space $\mathcal{X}$ and having task-specific output spaces $\{\mathcal{Y}_t\}_{t=1}^K$. We denote the training dataset by $\mathcal{D} = \{(x_i, \{y_{t,i}\}_{t=1}^K)\}_{i=1}^N$, drawn from a joint distribution $P(X, Y_1, \ldots, Y_K)$. A shared encoder $\Phi_\theta : \mathcal{X} \to \mathbb{R}^d$ with parameters $\theta$ maps an input $x$ to a shared representation $h = \Phi_\theta(x)$, which is then consumed by task-specific heads $\{f_{\phi_t}\}_{t=1}^K$ to produce predictions $h_t(x) = f_{\phi_t}(h)$.

Given per-task loss functions $\mathcal{L}_t$, the standard MTL objective on a source domain $S$ is the weighted risk

$$\min_{\theta, \{\phi_t\}} \sum_{t=1}^K w_t \, \mathbb{E}_{(x,y_t)\sim S}\big[\mathcal{L}_t\big(h_t(x), y_t\big)\big], \qquad (1)$$

where $w_t \geq 0$ denote task weights.

To study robustness, we distinguish between a source domain $S$ and a target domain $T$. The population risk of task $t$ on a domain $D \in \{S, T\}$ is

$$\mathcal{E}_D(h_t) = \mathbb{E}_{(x,y_t)\sim D}\Big[\mathcal{L}_t\big(h_t(x), y_t\big)\Big], \qquad (2)$$

and the domain gap of task $t$ is

$$\Delta_t(h) = \mathcal{E}_T(h_t) - \mathcal{E}_S(h_t). \qquad (3)$$

In the following sections, we analyze a lower bound on this gap $\Delta_t(h)$, aiming to identify structural properties of the shared representation $h$ that prevent standard multi-task training schemes from generalizing to target domains with style shifts.

### 2.2. Optimization-Centric MTL: An OOD Impossibility Result

To derive this limitation rigorously, we first formalize optimization-centric algorithms that dominate current MTL practice.

**Definition 2.1** (Gradient-balancing methods $\mathcal{G}$). Let $\Phi_\theta$ denote the shared encoder parameterized by $\theta$. A multi-task optimization algorithm belongs to $\mathcal{G}$ if it updates $\theta$ using directions of the form

$$g(\theta) = \sum_{t=1}^K w_t(\theta) \, \nabla_\theta \mathcal{L}_t(\theta), \qquad (4)$$

where the weights $w_t(\theta)$ are allowed to depend on the current gradients $\{\nabla_\theta \mathcal{L}_t(\theta)\}_t$, but the algorithm does not impose any structural constraint on the representation $h$.

In other words, methods such as GradNorm, PCGrad and MGDA may differ in how they choose $w_t(\theta)$, but all operate purely in gradient space and treat the shared feature geometry as fixed.

To expose the OOD limitation of $\mathcal{G}$, we adopt a simple causal model in which inputs are generated from invariant semantic factors and spurious residual factors. Let $Z_s$ denote invariant semantics and $Z_r$ denote residual factors capturing nuisance variation such as background or illumination, and assume at the population level

$$Z_s \perp Z_r, \qquad (5)$$

with observations obtained via a generative mechanism $X = g(Z_s, Z_r)$.

We now specialize this causal view to a linear-Gaussian structural causal model in which the encoder mixes $Z_s$ and $Z_r$ by a fixed rotation.

**Assumption 2.2** (Linear-Gaussian SCM with entanglement). Let $Z_s \sim \mathcal{N}(0, \Sigma_s)$ and $Z_r \sim \mathcal{N}(0, \Sigma_r^S)$ be independent latent semantic and residual factors. The encoder learns a representation

$$\hat{Z} = \Phi_\psi(X) = R_\psi[Z_s; Z_r], \qquad (6)$$

where $R_\psi$ is a rotation matrix with angle $\psi$ measuring how strongly $Z_s$ and $Z_r$ are mixed (Locatello et al., 2019). A domain shift changes only the residual covariance, from $\Sigma_r^S$ (source) to $\Sigma_r^T$ (target), while the distribution of $Z_s$ remains fixed. We adopt this standard invariant semantic assumption (Arjovsky et al., 2019) to focus our analysis on spurious correlations induced by nuisance shifts $\Sigma_r$, isolating them from label distribution shifts.

**Remark 1. (Inevitable Spurious Reliance)** Under Assumption 2.2, $\psi \neq 0$ implies that $\hat{Z}$ structurally mixes semantics with noise, forcing any downstream predictor to *incorporate* nuisance variations alongside robust features.

With this structural setup, we can show that any algorithm in $\mathcal{G}$ faces a non-vanishing OOD error floor whenever the representation remains entangled.

**Theorem 2.3** (OOD lower bound for gradient balancing). *Under Assumption 2.2, for any optimization-centric algorithm $A \in \mathcal{G}$, there exists a linear task (which is L-Lipschitz) such that if the learned representation remains entangled, i.e., $\psi \neq 0$, there exists a target domain shift where the generalization gap is lower-bounded by*

$$\mathcal{E}_T(h) - \mathcal{E}_S(h) \geq c \sin^2(\psi) \left\| \Sigma_r^T - \Sigma_r^S \right\|_F, \qquad (7)$$

*where $c > 0$ is a constant depending on the task geometry.*

*Proof.* See Appendix A. $\qquad\qquad\square$

**Remark 2. (Irreducible OOD Error Floor)** This lower bound establishes that no amount of gradient balancing can eliminate the $\sin^2 \psi$-scaled error; robustness is fundamentally limited by representation geometry, not optimization dynamics.

### 2.3. A Representation-Level Solution: Disentangled Architectures and Guarantees

The lower bound above suggests that robustness cannot be achieved by manipulating gradients alone; we must change the structure of the shared representation itself. We therefore consider architectures that explicitly disentangle semantic and residual streams and restrict task heads to depend only on the semantic stream. Formally, we factorize the encoder output as $(\hat{Z}_s, \hat{Z}_r) = \Phi_\theta(X)$, where $\hat{Z}_s$ is a semantic stream intended to capture invariant, task-relevant structure and $\hat{Z}_r$ is a residual stream that absorbs nuisance variation treated as task-irrelevant noise. Task predictions are built only from $\hat{Z}_s$, i.e., $h_t(x) = f_{\phi_t}(\hat{Z}_s)$.

To guarantee robustness, we rely on the structural properties of the learned representation. We model the semantic stream as a function $\hat{Z}_s = g_s(Z_s, Z_r)$ and define the leakage coefficient $\lambda_{\text{leak}}$ as the partial Lipschitz constant of $g_s$ with respect to the residual factors $Z_r$:

$$\lambda_{\text{leak}} \triangleq \sup_{z_s} \sup_{z_r \neq z_r'} \frac{\|g_s(z_s, z_r) - g_s(z_s, z_r')\|}{\|z_r - z_r'\|}. \qquad (8)$$

A value of $\lambda_{\text{leak}} = 0$ implies a structurally disentangled encoder where $\hat{Z}_s$ is invariant to $Z_r$. Under this definition, we derive the following bound.

**Theorem 2.4** (Disentangled OOD generalization bound). *Let $\mathcal{H}$ be the hypothesis class of models that predict solely from the semantic stream $\hat{Z}_s$. Under Assumption 2.2, there exist constants $C_{\text{cap}} \geq 0$ and $\alpha > 0$ such that for any $h \in \mathcal{H}$,*

$$\mathcal{E}_T(h) - \mathcal{E}_S(h) \leq C_{\text{cap}} + \alpha \lambda_{\text{leak}} W_1\big(P_S(Z_r), P_T(Z_r)\big), \qquad (9)$$

*where $W_1$ is the Wasserstein–1 distance between the source and target residual distributions.*

*Proof.* See Appendix C.

**Remark 3. (Linear Robustness Scaling)** Inequality (9) establishes that, under residual shifts, the OOD generalization gap is bounded by a term scaling linearly with the leakage coefficient $\lambda_{\text{leak}}$. This implies that geometric disentanglement is a fundamental prerequisite for robustness: suppressing the encoder's sensitivity to residual variations

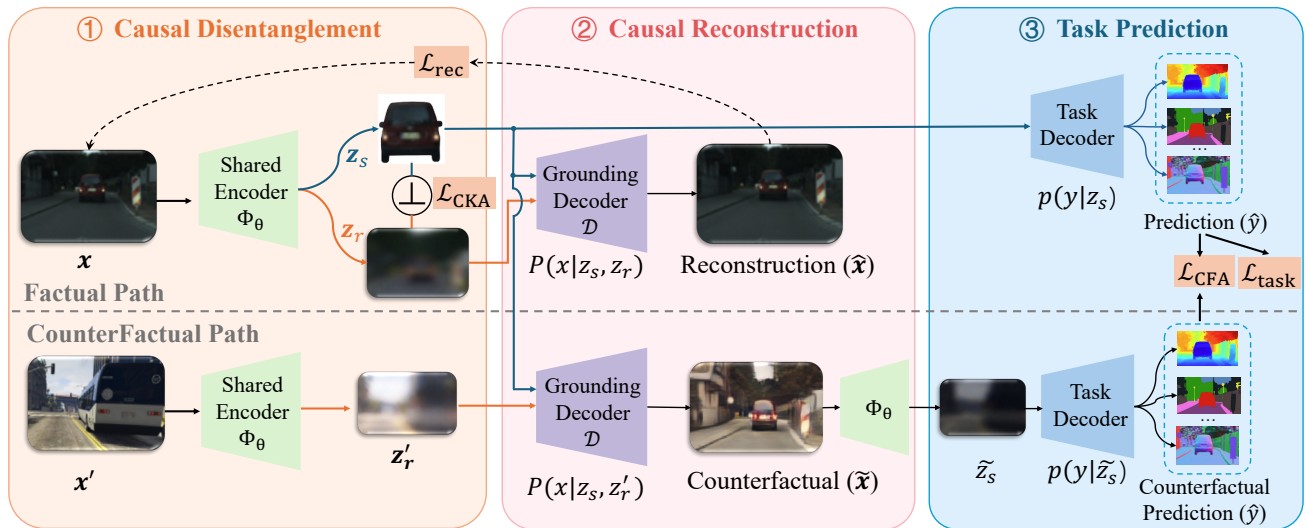

Figure 1. Overview of the CORE-MTL architecture.

serves to decouple the generalization bound from the magnitude of distribution shifts.

Beyond robustness, disentangling the shared representation also changes how task losses interact through the backbone. We analyze the interaction between the primary task gradient and the auxiliary residual gradient within the shared backbone $H$.

**Proposition 2.5** (Gradient Orthogonality via Disentanglement). *Consider a task loss $\mathcal{L}_{\text{task}}$ dependent on $Z_s$ and a residual regularization loss $\mathcal{L}_{\text{res}}$ dependent on $Z_r$. Let $g_{\text{task}}$ and $g_{\text{res}}$ be their gradients with respect to the shared representation $H$. Assuming the task heads have bounded gradients almost surely, there exist constants $c \geq 0$ and $\delta \geq 0$ such that:*

$$\mathbb{E}\left[\cos^2(g_{\text{task}}, g_{\text{res}})\right] \leq c \cdot \text{CKA}(Z_s, Z_r) + \delta, \quad (10)$$

*where $\text{CKA}(Z_s, Z_r)$ is the independence measure minimized during training.*

*Proof.* See Appendix D.

**Remark 4. (Structural Interference Mitigation)** Gradient orthogonality emerges as a structural by-product of disentanglement, resolving task conflicts at their geometric source rather than through post-hoc gradient surgery.

## 3. Method

Section 2 highlights that robustness under style shifts cannot be obtained by coordinating task gradients alone when the shared features remain entangled. To address this, we propose *CORE-MTL*, a representation-centric framework designed to structurally disentangle invariant semantics from nuisance variations. As illustrated in Figure 1, our method is composed of a generic meta-framework that enforces causal separation, and a grounding mechanism that anchors these factors via reconstruction.

### 3.1. The CORE-MTL Meta-Framework

Our meta-framework establishes the necessary structural conditions for disentanglement. It consists of three components: a dual-stream encoder, a structural independence constraint, and a counterfactual invariance mechanism.

**Dual-Stream Encoder.** We encode an input $x \in \mathcal{X}$ with a shared encoder $\Phi_\theta$ and explicitly factorize its output into two streams $(\hat{Z}_s, \hat{Z}_r) = \Phi_\theta(x)$, where $\hat{Z}_s$ is designated to capture shared task-relevant semantics (e.g., geometry, object identity) and $\hat{Z}_r$ absorbs complementary residual variation (e.g., texture, lighting, background). Crucially, task predictions are restricted to depend only on the semantic stream, $\hat{y}_t = f_{\phi_t}(\hat{Z}_s)$ for $t = 1, \ldots, K$. This structural constraint isolates task predictors from the residual stream, inherently preventing the model from exploiting nuisance variations as shortcuts.

**Structural Independence via CKA.** To encourage the semantic and residual representations to capture mutually exclusive information, we enforce statistical orthogonality. We use the linear Centered Kernel Alignment (CKA) (Kornblith et al., 2019) as our independence regularizer and define the independence loss as:

$$\mathcal{L}_{\text{CKA}} = \text{CKA}(\mathbf{Z}_s, \mathbf{Z}_r), \quad (11)$$

where $\mathbf{Z}_s, \mathbf{Z}_r$ are the feature matrices of the current mini-batch. This objective minimizes the linear dependence between the two streams. Crucially, under our linear–Gaussian

assumption, reducing this statistical dependence suppresses the Jacobian cross-terms, offering a tractable proxy for controlling the geometric leakage coefficient $\lambda_{\text{leak}}$ appearing in Theorem 2.4 (see Appendix D for a formal connection).

**Counterfactual Feature Augmentation.** To encourage the semantic representation $\hat{Z}_s$ to be invariant to nuisance factors, we introduce a **C**ounter**f**actual **A**ugmentation (CFA) mechanism based on counterfactual-style recombination. We instantiate an auxiliary decoder $\hat{x} = \mathcal{D}(\hat{Z}_s, \hat{Z}_r)$ to reconstruct the input.

During training, we synthesize counterfactual instances by intervening on the residual stream: we replace the original residual features with a random nuisance vector $\tilde{Z}_r$ drawn from the empirical residual distribution, generating a counterfactual image $\tilde{x} = \mathcal{D}(\hat{Z}_s, \tilde{Z}_r)$.

Crucially, since $\tilde{x}$ retains the original semantic content $\hat{Z}_s$ but possesses a different nuisance context, a robust encoder must still yield correct task predictions from it. We enforce this by passing the counterfactual image $\tilde{x}$ through the shared encoder and minimizing the consistency loss:

$$\mathcal{L}_{\text{CFA}} = \sum_{t=1}^{K} w_t \, \mathcal{L}_t \big( f_{\phi_t}([\Phi_\theta(\tilde{x})]_s), y_t \big), \quad (12)$$

where $[\Phi_\theta(\cdot)]_s$ denotes the semantic stream output of the shared encoder. In Figure 1, $\tilde{z}_s = [\Phi_\theta(\tilde{x})]_s$ denotes the semantic representation re-encoded from the synthesized image $\tilde{x}$. This mechanism discourages the task predictors from relying on spurious correlations between task labels and nuisance variations. Since $\mathcal{D}$ only enforces structural constraints during training, it imposes no computational overhead at inference time.

## 3.2. Grounding Disentanglement via Reconstruction

While the meta-framework enforces statistical independence, independence alone is insufficient to determine a semantically meaningful factor-role assignment, leaving the factorization potentially ambiguous. Without additional constraints, the model might satisfy the orthogonality objective by splitting features arbitrarily. To mitigate this ambiguity, we employ a generative reconstruction task as a proxy to ground the factorization. By minimizing a reconstruction objective ($\mathcal{L}_{\text{rec}}$), we encourage the latent factors to retain the necessary information to reproduce the input, thereby providing a grounding signal for assigning physical or semantic realities via the decoder $\mathcal{D}$. We instantiate the decoder $\mathcal{D}$ differently depending on the availability of strong inductive biases in the target tasks.

### 3.2.1. HARD GROUNDING VIA EXPLICIT PHYSICAL DECOMPOSITION

In the context of geometric scene understanding, we exploit the physical laws of light transport as a structural prior to encourage a grounded semantic–residual decomposition.

We formalize the grounding decoder $\mathcal{D}$ as a Physics-Based Inverse Rendering Decoder (architectural details in Appendix E). Specifically, unlike a generic decoder over concatenated latents, $D_{\text{phy}}$ uses role-specific heads: the semantic stream $\hat{Z}_s$ predicts geometric structure such as surface normals, while the residual stream $\hat{Z}_r$ predicts photometric factors such as albedo and illumination. These factors are composed through a fixed physical shading model, imposing an asymmetric geometry–photometry role assignment during reconstruction. The reconstruction is then synthesized via a physical shading model:

$$\hat{x} = \mathcal{D}_{\text{phy}}(\hat{Z}_s, \hat{Z}_r) \approx \underbrace{\mathcal{A}(\hat{Z}_r)}_{\text{Albedo}} \odot \underbrace{\mathcal{S}(\mathcal{N}(\hat{Z}_s), \mathbf{L}(\hat{Z}_r))}_{\text{Shading}}, \quad (13)$$

where $\mathcal{A}, \mathcal{N}, \mathbf{L}$ denote albedo, normals, and lighting, respectively, $\mathcal{S}$ is the shading function, and $\odot$ is the element-wise product. To align the latent factors with these physical quantities, we minimize a composite reconstruction objective:

$$\mathcal{L}_{\text{rec}} = \|x - \hat{x}\|_1 + \lambda_{\text{lpips}}\text{LPIPS}(x, \hat{x}). \quad (14)$$

where LPIPS refers to the Learned Perceptual Image Patch Similarity metric (Zhang et al., 2018). This hard grounding physically biases the geometry-focused representation away from texture-related nuisances to benefit downstream tasks.

### 3.2.2. SOFT GROUNDING VIA IMPLICIT ARCHITECTURAL CONSTRAINTS

For tasks where explicit physical equations are unavailable or overly complex, we demonstrate that CORE-MTL remains effective using soft grounding.

In this setting, we instantiate $\mathcal{D}$ as a Generic Convolutional Decoder ($\mathcal{D}_{\text{gen}}$) without explicit physical rendering equations. The grounding objective simplifies to a standard image reconstruction loss:

$$\mathcal{L}_{\text{rec}} = \|x - \mathcal{D}_{\text{gen}}(\hat{Z}_s, \hat{Z}_r)\|_1. \quad (15)$$

Here, a functional separation is encouraged from the architectural bottleneck and statistical independence. Since task heads only access $\hat{Z}_s$, discriminative features concentrate there, while the independence constraint ($\mathcal{L}_{\text{CKA}}$) forces remaining variations required for reconstruction into $\hat{Z}_r$. This mechanism functionally mirrors the explicit physical decomposition: while hard grounding uses explicit physical priors to provide stronger factor-role anchoring, soft grounding achieves a similar structural orthogonality through statistical pressure.

### 3.3. Optimization Objectives

The total objective function integrates the task supervision with the framework's regularization terms:

$$\mathcal{L}_{\text{total}} = \sum_{t=1}^{K} w_t \mathcal{L}_t + \lambda_{\text{CKA}}\, \mathcal{L}_{\text{CKA}} + \lambda_{\text{CFA}}\, \mathcal{L}_{\text{CFA}} + \lambda_{\text{rec}}\, \mathcal{L}_{\text{rec}},$$
(16)

where $\mathcal{L}_{\text{rec}}$ switches between Eq. (14) and Eq. (15) depending on the instantiation. During the counterfactual pass (Eq. (12)), we freeze Batch Normalization statistics to strictly evaluate robustness against synthesized style shifts.

## 4. Experiments

### 4.1. Experimental Setup

We evaluate on **NYUv2** (Silberman et al., 2012) and **Cityscapes** (Cordts et al., 2016) (in-distribution, ID) implemented with LibMTL (Lin & Zhang, 2023), **CelebA** (Liu et al., 2015) (scalability), **GTA5** (Richter et al., 2016)→ **Cityscapes**, and **Cityscapes**→ **Cityscapes-C** (Hendrycks & Dietterich, 2019; Michaelis et al., 2019) (out-of-distribution, OOD). We compare CORE-MTL with representative baselines spanning architecture-centric sharing (MTAN (Liu et al., 2019)), weighting or balancing (EW (Caruana, 1997), GradNorm (Chen et al., 2018), RLW (Lin et al., 2022), ExcessMTL (He et al., 2024)), gradient manipulation or fair allocation (PCGrad (Yu et al., 2020), Fair-Grad (Ban & Ji, 2024)), scalarization-based multi-objective optimization (STCH (Lin et al., 2024)), meta-optimization (MOML (Ye et al., 2021)), and representation-level methods (RepMTL (Wang et al., 2025)) using the same ResNet-50 backbone (He et al., 2016) unless otherwise specified. Architectural details are provided in Appendix E, while optimization and hyperparameters are deferred to Appendix F.

### 4.2. In-Distribution Results

On NYUv2 and Cityscapes, we evaluate semantic segmentation, depth estimation, and surface normal prediction using the standard metrics in Table 1. With the expanded comparison set, CORE-MTL achieves the strongest overall trade-off across datasets, obtaining the best segmentation and depth performance on NYUv2 and the best segmentation and depth relative error on Cityscapes, while remaining competitive on the remaining metrics. These results suggest that encouraging semantic–residual factorization improves dense multi-task prediction by reducing representational entanglement rather than relying solely on gradient reweighting or projection.

### 4.3. Out-of-Distribution Results

Table 2 evaluates robustness under sim-to-real transfer and common corruptions. On GTA5→Cityscapes, CORE-MTL

achieves the best target-domain segmentation, pixel accuracy, depth absolute error, and depth relative error among the compared methods, while also yielding the smallest mIoU and depth error gaps in most cases. This indicates that the learned semantic stream is less sensitive to source-specific residual variation under domain shift. On Cityscapes-C, CORE-MTL again achieves the best mIoU, pixel accuracy, and relative depth error, showing that the same factorization remains robust under corruption shifts. Overall, these results are consistent with Theorem 2.4: reducing residual leakage in the prediction stream improves OOD robustness.

### 4.4. Analysis of Structural Factorization

#### 4.4.1. ANALYSIS OF LATENT FACTORIZATION

**Structural factorization yields stable semantics and absorbs residual variations.** Qualitatively, Fig. 2 demonstrates that the counterfactual recombination $D(Z_s^{\text{CS}}, Z_r^{\text{GTA}})$ preserves scene layout while altering appearance statistics, supporting the intended factorization. Quantitatively, the feature-swapping analysis (Fig. 3, Table 3) reveals a high robustness ratio ($\Delta Z_r / \Delta Z_s$ significantly larger than 1), confirming that $Z_s$ remains stable under domain shifts while $Z_r$ absorbs variations. Together, these results demonstrate that the learned structure prioritizes task-relevant semantics and buffers domain-specific nuisances, explaining the downstream gains in robustness.

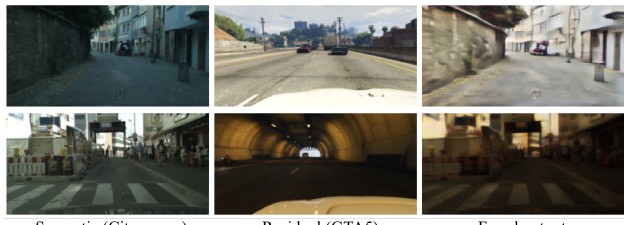

Semantic (Cityscapes)    Residual (GTA5)    Fused output

*Figure 2.* **Cross-domain feature swapping.** Left: Cityscapes content (semantic stream). Middle: GTA5 reference (residual stream). Right: Synthesized output combining Cityscapes semantics with GTA5 residual appearance.

**Colored-Cityscapes stress test.** Since Assumption 2.2 is idealized while real scenes often exhibit strong semantic–context correlations, we construct a shortcut-based Colored-Cityscapes benchmark: class colors are fixed in the source domain but permuted in the target domain. Appendix I shows that CORE-MTL remains the strongest under this shortcut shift, suggesting that its semantic–residual factorization reduces residual-appearance leakage rather than relying on literal semantic–residual independence.

#### 4.4.2. ANALYSIS OF GRADIENT ORTHOGONALITY

**Structure induces intrinsic gradient orthogonality.** Unlike explicit gradient-projection methods such as PCGrad, Fig. 4 shows that CORE-MTL achieves near-zero cosine

*Table 1.* **In-Distribution performance.** Left: NYUv2 (3 tasks). Right: Cityscapes (2 tasks). ↑ / ↓ indicate higher / lower is better.

| Method | NYUv2 (3 Tasks) | | | | | | | | | Cityscapes (2 Tasks) | | | |
|---|---|---|---|---|---|---|---|---|---|---|---|---|---|
| | Segmentation | | Depth | | Surface Normal | | | | | Segmentation | | Depth | |
| | mIoU↑ | Pix Acc↑ | Abs Err↓ | Rel Err↓ | Mean↓ | Med↓ | 11.25°↑ | 22.5°↑ | 30°↑ | mIoU↑ | Pix Acc↑ | Abs Err↓ | Rel Err↓ |
| Single Task | 0.5192 | 0.7396 | 0.5260 | 0.2076 | 24.2676 | 18.6778 | 0.3071 | 0.5800 | 0.7048 | 0.6869 | 0.9144 | 0.0129 | 47.9603 |
| Equal Weighting | 0.5316 | 0.7508 | 0.3911 | 0.1633 | 24.2909 | 17.7635 | 0.3393 | 0.5927 | 0.7051 | 0.6962 | 0.9192 | 0.0128 | 44.0340 |
| PCGrad | 0.5222 | 0.7433 | 0.3916 | 0.1609 | 24.3858 | 17.7999 | 0.3398 | 0.5915 | 0.7039 | 0.6998 | 0.9187 | 0.0128 | 44.5574 |
| GradNorm | 0.5264 | 0.7436 | 0.3896 | 0.1648 | 24.3864 | 17.6391 | 0.3425 | 0.5944 | 0.7055 | 0.7015 | 0.9197 | 0.0125 | 44.5520 |
| STCH | 0.5377 | 0.7483 | 0.3917 | 0.1569 | 23.2045 | 16.5197 | 0.3610 | 0.6203 | 0.7286 | 0.6952 | 0.9180 | 0.0123 | 42.8384 |
| MTAN | 0.5401 | 0.7542 | 0.3822 | 0.1610 | 24.0165 | 17.3356 | 0.3462 | 0.6023 | 0.7130 | 0.7023 | 0.9206 | 0.0126 | 45.5663 |
| MOML | 0.5303 | 0.7496 | 0.3952 | 0.1612 | 24.0679 | 17.4130 | 0.3469 | 0.5987 | 0.7093 | 0.6876 | 0.9146 | 0.0145 | 52.5924 |
| RLW | 0.5293 | 0.7510 | 0.3883 | 0.1601 | 24.0588 | 17.2890 | 0.3465 | 0.6033 | 0.7134 | 0.6995 | 0.9185 | 0.0131 | 43.6991 |
| ExcessMTL | 0.4846 | 0.7180 | 0.3877 | 0.1624 | 26.1446 | 19.8459 | 0.3043 | 0.5499 | 0.6676 | 0.6969 | 0.9195 | 0.0128 | 43.3338 |
| FairGrad | 0.5291 | 0.7427 | 0.3944 | 0.1585 | 23.0717 | **16.3569** | **0.3657** | **0.6239** | 0.7312 | 0.6986 | 0.9194 | **0.0121** | 43.7426 |
| RepMTL | 0.5492 | 0.7598 | 0.3727 | 0.1503 | 24.5348 | 19.6139 | 0.3018 | 0.5576 | 0.6847 | 0.7079 | 0.9184 | 0.0180 | 44.3230 |
| **CORE-MTL (Ours)** | **0.5693** | **0.7759** | **0.3544** | **0.1466** | **22.4927** | 16.8337 | 0.3501 | 0.6196 | **0.7360** | **0.7229** | **0.9245** | 0.0123 | **19.6088** |

*Table 2.* **Out-of-distribution generalization.** Left: Sim-to-real GTA5→Cityscapes with source, target, and gap Δ (performance drop or error gap). Right: Cityscapes-C robustness averaged over four corruptions (per-corruption results in Appendix J).

| Method | GTA5 → Cityscapes (Sim-to-Real) | | | | | | | | | | | | Cityscapes-C | | | |
|---|---|---|---|---|---|---|---|---|---|---|---|---|---|---|---|---|
| | mIoU↑ | | | Pixel Acc↑ | | | Abs Err↓ | | | Rel Err↓ | | | mIoU↑ | Pixel Acc↑ | Abs Err↓ | Rel Err↓ |
| | Source | Target | Δ↓ | Source | Target | Δ↓ | Source | Target | Δ↓ | Source | Target | Δ↓ | | | | |
| Equal Weighting | 0.6611 | 0.5407 | 0.1204 | 0.9347 | 0.8244 | 0.1103 | 0.0173 | 0.1347 | 0.1174 | 0.2675 | 303.07 | 302.80 | 0.5831 | 0.8521 | 0.0239 | 58.75 |
| PCGrad | 0.6605 | 0.5047 | 0.1558 | 0.9344 | 0.7943 | 0.1401 | 0.0162 | 0.1435 | 0.1273 | 0.2472 | 301.85 | 301.61 | 0.5851 | 0.8520 | 0.0230 | 59.20 |
| GradNorm | 0.6590 | 0.5233 | 0.1357 | 0.9352 | 0.8112 | 0.1240 | 0.0161 | 0.1364 | 0.1203 | 0.2062 | 294.48 | 294.27 | 0.5900 | 0.8522 | 0.0213 | 59.04 |
| STCH | 0.6548 | 0.5177 | 0.1371 | 0.9334 | 0.8240 | 0.1094 | **0.0158** | 0.1712 | 0.1554 | **0.1542** | 333.10 | 332.95 | 0.5853 | 0.8537 | 0.0192 | 54.57 |
| MTAN | 0.6637 | 0.5147 | 0.1490 | 0.9366 | 0.8016 | 0.1350 | 0.0163 | 0.1185 | 0.1022 | 0.1905 | 272.52 | 272.33 | 0.5887 | 0.8549 | 0.0218 | 56.55 |
| MOML | 0.6548 | 0.5314 | 0.1234 | 0.9417 | 0.8323 | 0.1094 | 0.0173 | 0.1235 | 0.1062 | 0.9818 | 287.25 | 286.27 | 0.5989 | 0.8633 | 0.0239 | 54.01 |
| RLW | 0.6629 | 0.5203 | 0.1426 | 0.9336 | 0.8119 | 0.1217 | 0.0177 | 0.1406 | 0.1229 | 0.1916 | 309.51 | 309.32 | 0.5916 | 0.8582 | 0.0212 | 56.82 |
| ExcessMTL | 0.6587 | 0.5430 | 0.1157 | 0.9361 | 0.8382 | **0.0979** | 0.0167 | 0.1586 | 0.1419 | 0.1895 | 321.17 | 320.98 | 0.5814 | 0.8534 | 0.0233 | 60.28 |
| FairGrad | **0.6665** | 0.4922 | 0.1743 | 0.9343 | 0.7756 | 0.1587 | 0.0266 | 0.1522 | 0.1256 | 0.5264 | 320.55 | 320.02 | 0.5806 | 0.8490 | 0.0188 | 54.46 |
| RepMTL | 0.6457 | 0.5343 | 0.1115 | 0.9416 | 0.8120 | 0.1313 | 0.0211 | 0.1128 | 0.0916 | 0.8887 | 277.73 | 276.84 | 0.5916 | 0.8560 | **0.0181** | 51.50 |
| **CORE-MTL (Ours)** | 0.6500 | **0.5435** | **0.1065** | **0.9422** | **0.8401** | 0.1021 | 0.0220 | **0.0979** | **0.0759** | 0.1882 | **235.23** | **235.04** | **0.6104** | **0.8670** | 0.0182 | **38.10** |

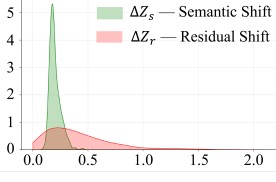

*(a)* Intra-domain stability

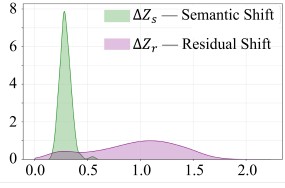

*(b)* Cross-domain stability

*Figure 3.* Stability analysis of semantic ($Z_s$) and residual ($Z_r$) streams. We measure feature shifts ($\Delta Z$) under (a) intra-domain residual shuffling and (b) cross-domain intervention.

*Table 3.* Stability analysis under interventions. We report feature shifts ($\Delta Z$) and the Robustness Ratio ($\Delta Z_r / \Delta Z_s$). High ratio indicates effective disentanglement.

| Setting | $\Delta Z_s$(Sem) ↓ | $\Delta Z_r$(Res) ↑ | Ratio ↑ |
|---|---|---|---|
| Intra-Domain | 0.1997 | 0.4158 | 2.08 |
| Cross-Domain | 0.2889 | 0.9466 | **3.28** |

similarity between task and reconstruction gradients as a geometric by-product of semantic–residual factorization. Fig. 5 instead visualizes pairwise task–task gradient interactions, where the structured block patterns indicate more organized task relationships rather than uniformly vanishing gradients. Thus, resolving interference at the representation

level reshapes gradient interactions and reduces the reliance on post-hoc gradient surgery.

### 4.5. Scalability Results

**Structural orthogonality offers superior scalability.** As shown in Table 4, CORE-MTL maintains near-constant training cost as the number of tasks $K$ increases, avoiding the prohibitive linear scaling of gradient surgery (PCGrad

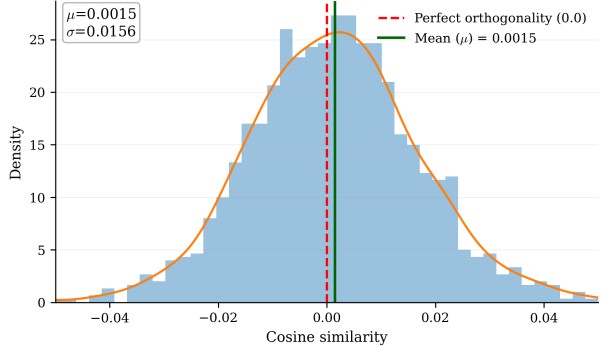

*Figure 4.* Subspace orthogonality on CelebA. Distribution of cosine similarities between attribute task gradients and reconstruction gradients at the last shared encoder layer.

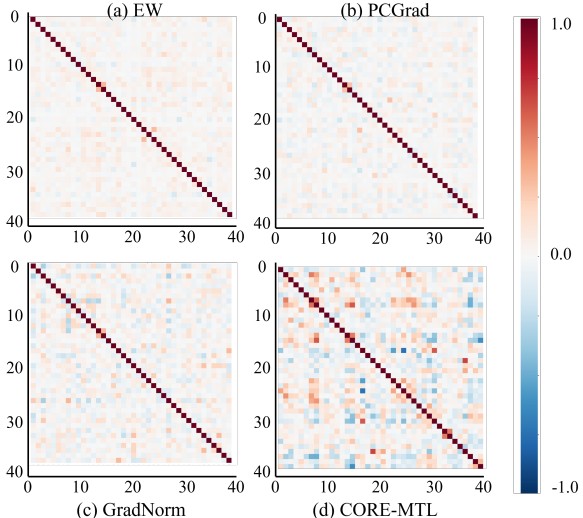

*Figure 5.* **Gradient interaction heatmaps on CelebA.** Pairwise cosine similarities of task gradients at the last shared layer (see Appendix K for the attribute order).

*Table 4.* **Scalability analysis on CelebA.** Training time per epoch and average attribute accuracy vs. number of tasks ($K$).

| #Tasks $K$ | Method | Time / epoch (s)↓ | Avg Accuracy (%)↑ |
|---|---|---|---|
| 10 | Equal Weighting (EW) | 86 | 88.52 |
| | PCGrad | 690 | 88.32 |
| | STCH | 89 | 88.39 |
| | **CORE-MTL (Ours)** | 297 | **89.39** |
| 20 | Equal Weighting (EW) | 91 | 91.55 |
| | PCGrad | 1396 | 91.23 |
| | STCH | 88 | 90.81 |
| | **CORE-MTL (Ours)** | 297 | **91.80** |
| 30 | Equal Weighting (EW) | 92 | 91.40 |
| | PCGrad | 1950 | 91.41 |
| | STCH | 91 | 90.80 |
| | **CORE-MTL (Ours)** | 297 | **91.76** |
| 40 | Equal Weighting (EW) | 94 | 91.52 |
| | PCGrad | 2806 | 91.34 |
| | STCH | 97 | 91.01 |
| | **CORE-MTL (Ours)** | 300 | **91.77** |

reaches 2806 s/epoch at $K{=}40$). Although the auxiliary decoder incurs a fixed overhead ($\approx 300$ s/epoch) compared to simple weighting ($\approx 90$ s/epoch), this cost does not compound with the number of tasks. Notably, CORE-MTL yields the highest average accuracy across different numbers of tasks, demonstrating a superior trade-off that avoids the prohibitive scaling of gradient surgery while consistently outperforming naive baselines.

### 4.6. Ablation Studies

#### 4.6.1. COMPONENT ABLATION

Table 5 shows that introducing the Dual-Stream (DS) backbone already improves over Vanilla MTL, confirming the benefit of separating semantics from noise. Adding the auxiliary reconstruction task yields further gains by providing an additional grounding signal that stabilizes the dual-stream decomposition and discourages shortcut solutions. The full model (DS+Grounding+CKA+CFA) achieves the strongest

overall performance, indicating that CKA and CFA are complementary: CKA encourages cross-stream independence (Theorem 2.4), while CFA encourages prediction consistency under counterfactual-style residual substitution and reduces reliance on spurious correlations.

#### 4.6.2. GROUNDING DECODER DESIGN

We further examine whether the effectiveness of CORE-MTL relies on the explicit physics-based decoder. To this end, we replace the hard physical grounding decoder with a generic convolutional decoder while keeping the semantic–residual factorization and training objectives unchanged. As shown in Table 6, soft grounding already achieves competitive performance, indicating that CORE-MTL does not solely rely on hand-crafted physical rendering equations. Meanwhile, hard grounding obtains slightly stronger overall performance, suggesting that physical priors provide a useful inductive bias for dense scene understanding.

#### 4.6.3. SENSITIVITY TO INDEPENDENCE REGULARIZATION

**Sensitivity analysis reveals the independence-capacity trade-off.** We further investigate the impact of the independence constraint weight $\lambda_{\mathrm{CKA}}$ and the counterfactual augmentation weight $\lambda_{\mathrm{CFA}}$. Detailed analyses for the independence regularization are provided in Appendix G, while the sensitivity to counterfactual perturbation magnitude is analyzed in Appendix H. In summary, we observe a clear "sweet spot" for $\lambda_{\mathrm{CKA}}$ (e.g., 1.0 on NYUv2). Overly weak regularization allows leakage, while excessive regularization constrains the semantic stream's capacity. Similarly, the CFA ablation in Appendix H confirms that moderate counterfactual perturbations effectively regularize the model against OOD shifts, whereas excessive perturbation noise can destabilize the learning of the semantic manifold.

### 5. Related Work

**Optimization-centric multi-task optimization.** Standard approaches focus on weighting or balancing losses (e.g., GradNorm (Chen et al., 2018), uncertainty (Kendall et al., 2018), RLW (Lin et al., 2022), ExcessMTL (He et al., 2024)) or pursuing multi-objective optimization (MGDA (Sener & Koltun, 2018), MOML (Ye et al., 2021), STCH (Lin et al., 2024)). To mitigate conflicting gradients, "surgery" or allocation-based methods modify updates (e.g., PCGrad (Yu et al., 2020), CAGrad (Liu et al., 2021), Nash-MTL (Navon et al., 2022), FairGrad (Ban & Ji, 2024)), while recent works refine conflict geometry via rotation (Javaloy & Valera, 2022) or conic constraints (Hassanpour et al., 2025). Strategies like accumulation-based stabilization (Limarenko & Studenikina, 2025), Dual-Balancing (Lin et al., 2025), AutoScale (Yang et al., 2025), and MLB (Kontras et al.,

*Table 5.* **Component-wise ablation on NYUv2.** DS: Dual Stream; Grounding: Grounding with Reconstruction; CKA: Independence Loss; CFA: Counterfactual Augmentation.

| | Components | | | | Segmentation | | Depth | | Surface Normal | | | | |
|---|---|---|---|---|---|---|---|---|---|---|---|---|---|
| Method | Dual | Grounding | CKA | CFA | mIoU↑ | PixAcc↑ | Abs↓ | Rel↓ | Mean↓ | Med↓ | 11.25° ↑ | 22.5° ↑ | 30° ↑ |
| Vanilla MTL | | | | | 0.5249 | 0.7417 | 0.4418 | 0.1881 | 25.6201 | 19.8490 | 0.2926 | 0.5521 | 0.6776 |
| DS | ✓ | | | | 0.5352 | 0.7527 | 0.3813 | 0.1622 | 23.2973 | 16.9788 | **0.3517** | 0.6117 | 0.7230 |
| DS+Grounding | ✓ | ✓ | | | 0.5424 | 0.7546 | 0.3827 | 0.1611 | 23.2189 | 16.9394 | **0.3517** | 0.6130 | 0.7267 |
| DS+Grounding+CKA | ✓ | ✓ | ✓ | | 0.5547 | 0.7630 | 0.3876 | 0.1636 | 23.9179 | 18.5226 | 0.3165 | 0.5814 | 0.7041 |
| Full | ✓ | ✓ | ✓ | ✓ | **0.5693** | **0.7759** | **0.3544** | **0.1466** | **22.4927** | **16.8337** | 0.3501 | **0.6196** | **0.7360** |

*Table 6.* **Ablation study on grounding decoder design on NYUv2.** Soft grounding uses a generic convolutional decoder, while hard grounding uses the physics-based grounding decoder.

| Setting | Segmentation | | Depth | | Surface Normal | | | | |
|---|---|---|---|---|---|---|---|---|---|
| | mIoU↑ | Pix Acc↑ | Abs Err↓ | Rel Err↓ | Mean↓ | Med.↓ | 11.25° ↑ | 22.5° ↑ | 30° ↑ |
| Soft Grounding | 0.5652 | 0.7715 | 0.3555 | **0.1427** | 23.0044 | 17.3100 | 0.3415 | 0.6068 | 0.7238 |
| Hard Grounding | **0.5693** | **0.7759** | **0.3544** | 0.1466 | **22.4927** | **16.8337** | **0.3501** | **0.6196** | **0.7360** |

2024) further stabilize training, yet typically address symptoms rather than the root cause of representation entanglement (Shi et al., 2023).

**Architecture-centric parameter sharing.** Architectural methods reduce interference by modulating sharing. Approaches like MTAN (Liu et al., 2019) and affinity-based designs (Sinodinos & Armanfard, 2025) encourage selective feature reuse. Mixture-of-experts models (e.g., MMoE (Ma et al., 2018), PLE (Tang et al., 2020)) decouple tasks via gated experts, with recent variants extending to large-scale backbones (Kong et al., 2025; Mehta et al., 2025). However, routing decisions in these models remain primarily loss-driven rather than causal, leaving them susceptible to spurious correlations (Maheronnaghsh & Alvanagh, 2024).

**Causal and structured representation learning.** Robust generalization requires separating stable signals from nuisance factors. While subspace methods like DSN (Bousmalis et al., 2016), invariant frameworks (e.g., IRM (Arjovsky et al., 2019), MRI (Huh & Baidya, 2022)), and representation-level methods such as RepMTL (Wang et al., 2025) improve domain separation, invariance, or task saliency, we repurpose disentanglement as a structural route toward gradient orthogonality (Proposition 2.5), tackling a key optimization source of negative transfer. Similarly, unlike generative approaches utilizing feature-statistics mixing (MixStyle (Zhou et al., 2021)) for synthesis, we integrate counterfactual mechanisms as a regularizer to discourage reliance on spurious shortcuts (Sun et al., 2025). Crucially, distinct from causal frameworks that rely on modular routing to select invariant subnetworks (Hu et al., 2022), CORE-MTL encourages internal semantic–residual factorization via physical or generic grounding. This provides factor-role anchoring for the semantic–residual split and structurally induces gradient orthogonality as a geometric byproduct, mit-igating entanglement-induced transfer conflicts (Lachapelle et al., 2023; Hendawy et al., 2024; Tian et al., 2025).

# 6. Conclusion and Limitations

**Conclusion.** We showed that entangled shared representations, rather than the gradient balancing scheme itself, are a key source of task interference. Targeting this issue, CORE-MTL shifts the paradigm from post-hoc gradient surgery to intrinsic representation disentanglement. By imposing a causally motivated semantic–residual structural bias, our framework resolves interference at its source and achieves superior OOD robustness. More broadly, our findings advocate a shift from purely optimization-centric task coordination toward structure-aware representation design for robust multi-task learning.

**Limitations.** Our theoretical analysis relies on a stylized linear-Gaussian latent SCM, and extending the guarantees to deep nonlinear representations remains future work. CORE-MTL is instantiated mainly for dense scene understanding tasks; more heterogeneous or multimodal task sets may require more flexible grounding mechanisms.

# Impact Statement

CORE-MTL aims to improve out-of-distribution robustness in shared perception models by separating task-relevant semantic structure from nuisance variation. This may help mitigate dataset bias and distribution shift in applications such as scene understanding or assistive perception. However, as a general-purpose technique, it could also be used in sensitive settings such as surveillance or facial analysis, where fairness, privacy, and misuse concerns remain. Practitioners should conduct application-specific evaluations and safeguards before deployment in high-stakes settings.

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

## A. Proof of Theorem 2.3 (OOD lower bound for gradient balancing)

*Proof.* We give a concrete construction under the Linear-Gaussian SCM (Assumption 2.2) showing that if $\psi \neq 0$ (entanglement), then there exists a residual shift inducing a non-vanishing OOD gap. The argument is by exhibiting one task/predictor for which any optimization-centric method in $\mathcal{G}$ can converge on source risk yet still suffer a target shift through the entangled coordinate.

**Setup (a single linear task suffices).** Let $Z_s \sim \mathcal{N}(0,1)$ and $Z_r \sim \mathcal{N}(0, \sigma_{r,S}^2)$ be independent. Let the label be generated by

$$Y = w^\star Z_s + \varepsilon, \qquad \varepsilon \sim \mathcal{N}(0, \sigma^2), \tag{17}$$

and let the (entangled) scalar representation used by the head be

$$\hat{Z} = \cos \psi \, Z_s + \sin \psi \, Z_r. \tag{18}$$

Consider a linear head $h(\hat{Z}) = \beta \hat{Z}$ trained by minimizing source squared loss. This is a special case of our main setup with Lipschitz head; proving the lower bound here suffices because Theorem 2.3 is existential over target shifts and constants.

**Step 1: Source-optimal solution.** On the source domain, the population optimal coefficient is

$$\beta^\star = \frac{\text{Cov}_S(Y, \hat{Z})}{\text{Var}_S(\hat{Z})}. \tag{19}$$

Using (17)–(18) and independence $Z_s \perp Z_r$,

$$\text{Cov}_S(Y, \hat{Z}) = \mathbb{E}\big[(w^\star Z_s)(\cos \psi Z_s + \sin \psi Z_r)\big] = w^\star \cos \psi \, \mathbb{E}[Z_s^2] = w^\star \cos \psi, \tag{20}$$

$$\text{Var}_S(\hat{Z}) = \mathbb{E}\big[(\cos \psi Z_s + \sin \psi Z_r)^2\big] = \cos^2 \psi \, \text{Var}(Z_s) + \sin^2 \psi \, \text{Var}_S(Z_r) = \cos^2 \psi + \sin^2 \psi \, \sigma_{r,S}^2. \tag{21}$$

Hence

$$\beta^\star = \frac{w^\star \cos \psi}{\cos^2 \psi + \sin^2 \psi \, \sigma_{r,S}^2}. \tag{22}$$

**Step 2: Target risk under a residual covariance shift.** Now define a target domain by keeping $Z_s$ invariant but changing only the residual variance: $Z_r \sim \mathcal{N}(0, \sigma_{r,T}^2)$ with $\sigma_{r,T}^2 \neq \sigma_{r,S}^2$. The target risk of the source-trained predictor is

$$\mathcal{E}_T(h) - \mathcal{E}_S(h) = \mathbb{E}_T\big[(Y - \beta^\star \hat{Z})^2\big] - \mathbb{E}_S\big[(Y - \beta^\star \hat{Z})^2\big]. \tag{23}$$

Expanding $Y - \beta^\star \hat{Z}$ with (17)–(18) yields

$$Y - \beta^\star \hat{Z} = \big(w^\star - \beta^\star \cos \psi\big) Z_s - \beta^\star \sin \psi \, Z_r + \varepsilon. \tag{24}$$

Because $Z_s \perp Z_r \perp \varepsilon$, cross terms vanish in expectation, and the only term that changes between source and target is the variance contribution from $Z_r$:

$$\mathbb{E}_T[(Y - \beta^\star \hat{Z})^2] - \mathbb{E}_S[(Y - \beta^\star \hat{Z})^2] = (\beta^\star)^2 \sin^2 \psi \cdot \big(\text{Var}_T(Z_r) - \text{Var}_S(Z_r)\big) \tag{25}$$

$$= (\beta^\star)^2 \sin^2 \psi \cdot \big(\sigma_{r,T}^2 - \sigma_{r,S}^2\big). \tag{26}$$

**Step 3: Lower bound in the form of Theorem 2.3.** If $\psi \neq 0$, choose a target shift with $\sigma_{r,T}^2 > \sigma_{r,S}^2$, then (26) is strictly positive. Moreover, substituting (22) gives

$$\mathcal{E}_T(h) - \mathcal{E}_S(h) = \underbrace{\left( \frac{(w^\star)^2 \cos^2 \psi}{(\cos^2 \psi + \sin^2 \psi \, \sigma_{r,S}^2)^2} \right)}_{\triangleq c_0 > 0} \sin^2 \psi \cdot (\sigma_{r,T}^2 - \sigma_{r,S}^2). \tag{27}$$

This matches the claimed $\sin^2 \psi$ scaling. For the vector case in Assumption 2.2, take a diagonal residual covariance shift so that $\|\Sigma_r^T - \Sigma_r^S\|_F$ is proportional to the variance change along one coordinate, and absorb the proportionality into $c$. This construction establishes an OOD error floor in the simplest linear task setting. Since the constructed linear predictor $h(\hat{Z}) = \beta^* \hat{Z}$ is $L$-Lipschitz with $L = |\beta^*|$, this counterexample suffices to prove that entanglement alone induces unavoidable residual-driven errors for the class of Lipschitz functions, limiting any optimization-centric algorithm in $\mathcal{G}$. $\qquad \square$

## B. A concrete independence proxy (Linear CKA) and its link to entanglement

In the main text, $\mathrm{Indep}(\hat{Z}_s, \hat{Z}_r)$ is left as a generic dependence measure. In our implementation we use Linear CKA, and in this appendix we make explicit how it connects to the entanglement angle in Assumption 2.2.

**Linear CKA (batch form).**   Given centered batch feature matrices $U \in \mathbb{R}^{n \times d_u}$ and $V \in \mathbb{R}^{n \times d_v}$, the (linear-kernel) CKA is

$$\mathrm{CKA}(U, V) = \frac{\|U^\top V\|_F^2}{\|U^\top U\|_F \, \|V^\top V\|_F}. \tag{28}$$

When both are batch-standardized so that $\frac{1}{n} U^\top U \approx I$ and $\frac{1}{n} V^\top V \approx I$, Linear CKA is proportional to the squared Frobenius norm of the empirical cross-correlation matrix.

**Lemma B.1** (CKA as an entanglement proxy). *Assume batch-standardized features* $\mathrm{Cov}(\hat{Z}_s) = I$ *and* $\mathrm{Cov}(\hat{Z}_r) = I$ *(empirically, after centering/normalization). Then* $\mathrm{CKA}(\hat{Z}_s, \hat{Z}_r) = \| \mathrm{Corr}(\hat{Z}_s, \hat{Z}_r)\|_F^2$ *up to a constant factor.*

*Proof.* Under standardization, $\|\hat{Z}_s^\top \hat{Z}_s\|_F$ and $\|\hat{Z}_r^\top \hat{Z}_r\|_F$ are constants, so (28) reduces to $\|\hat{Z}_s^\top \hat{Z}_r\|_F^2$ up to scaling, i.e., squared cross-correlation. □

**A convenient choice of Indep.**   To align with Theorem 2.4, it is often convenient to take

$$\mathrm{Indep}(\hat{Z}_s, \hat{Z}_r) \; \triangleq \; \sqrt{\mathrm{CKA}(\hat{Z}_s, \hat{Z}_r)} \; \approx \; \| \mathrm{Corr}(\hat{Z}_s, \hat{Z}_r)\|_F, \tag{29}$$

so that the shift attenuation term is linear in the "leakage magnitude".

## C. Proof of Theorem 2.4

*Proof.* We derive a capacity-aware OOD bound where the effect of residual shift is attenuated by the leakage coefficient $\lambda_{\mathrm{leak}}$.

**Step 1: Standard Domain Adaptation Bound.** Let $\mathcal{H}$ be the hypothesis class of models predicting only from $\hat{Z}_s$. Standard learning-theoretic results (e.g., Wasserstein-based bounds for Lipschitz losses) bound the target risk by the source risk plus the distribution shift of the features:

$$\mathcal{E}_T(h) - \mathcal{E}_S(h) \leq C_{\mathrm{cap}} + \alpha_0 \, W_1\big(P_S(\hat{Z}_s), P_T(\hat{Z}_s)\big), \tag{30}$$

where $C_{\mathrm{cap}}$ handles capacity terms and $\alpha_0$ depends on the task head's Lipschitz constant.

**Step 2: Bounding Semantic Shift via Leakage.** Under Assumption 2.2, domain shift affects only $Z_r$. Thus, the joint distributions are $P_D(Z_s, Z_r) = P(Z_s) \otimes P_D(Z_r)$ for $D \in \{S, T\}$. Let $\pi_r$ be an optimal coupling between $P_S(Z_r)$ and $P_T(Z_r)$. We construct a joint coupling $\Pi$ by sampling $Z_s \sim P(Z_s)$ and $(Z_r^S, Z_r^T) \sim \pi_r$ independently. Using the definition of $\lambda_{\mathrm{leak}}$ as the partial Lipschitz constant:

$$\|\hat{Z}_s(Z_s, Z_r^S) - \hat{Z}_s(Z_s, Z_r^T)\| \leq \lambda_{\mathrm{leak}} \|Z_r^S - Z_r^T\|.$$

Taking expectations under the coupling $\Pi$ yields:

$$\begin{aligned}
W_1(P_S(\hat{Z}_s), P_T(\hat{Z}_s)) &\leq \mathbb{E}_\Pi[\|\hat{Z}_s^S - \hat{Z}_s^T\|] \\
&\leq \lambda_{\mathrm{leak}} \mathbb{E}_{\pi_r}[\|Z_r^S - Z_r^T\|] \\
&= \lambda_{\mathrm{leak}} W_1(P_S(Z_r), P_T(Z_r)).
\end{aligned} \tag{31}$$

**Conclusion.** Substituting (31) into (30) yields the claim. □

## D. Proof of Proposition 2.5

*Proof.* To establish the bound, we first define the Jacobians of the semantic and residual mappings with respect to the shared backbone $H$. Let $Z_s = \phi_s(H)$ and $Z_r = \phi_r(H)$, and denote their Jacobians by:

$$J_s = \frac{\partial Z_s}{\partial H}, \qquad J_r = \frac{\partial Z_r}{\partial H}. \tag{32}$$

By the chain rule, the gradients of the task and residual losses with respect to $H$ are:

$$g_{\text{task}} = J_s^\top v_s, \qquad g_{\text{res}} = J_r^\top v_r, \tag{33}$$

where $v_s = \nabla_{Z_s}\mathcal{L}_{\text{task}}$ and $v_r = \nabla_{Z_r}\mathcal{L}_{\text{res}}$ are the error signals backpropagated from the respective heads.

**Step 1: Bounding gradient cosine similarity.** We analyze the squared cosine similarity: $\cos^2(g_{\text{task}}, g_{\text{res}}) = (\langle g_{\text{task}}, g_{\text{res}}\rangle)^2/(\|g_{\text{task}}\|^2\|g_{\text{res}}\|^2)$. We explicitly expand the inner product using the Jacobian definitions:

$$\langle g_{\text{task}}, g_{\text{res}}\rangle = \langle J_s^\top v_s, J_r^\top v_r\rangle = v_s^\top (J_s J_r^\top) v_r. \tag{34}$$

Applying the Cauchy-Schwarz inequality:

$$|v_s^\top (J_s J_r^\top) v_r| \leq \|v_s\|\|v_r\|\|J_s J_r^\top\|_F. \tag{35}$$

Under the assumption that head gradients $\|v_s\|$ and $\|v_r\|$ are bounded almost surely (e.g., by gradient clipping or Lipschitz continuity of the heads) and that backbone gradients are non-degenerate, there exist constants $C_1, \delta_1 \geq 0$ such that:

$$\cos^2(g_{\text{task}}, g_{\text{res}}) \leq C_1\|J_s J_r^\top\|_F^2 + \delta_1. \tag{36}$$

Taking the expectation over the data distribution yields:

$$\mathbb{E}[\cos^2(g_{\text{task}}, g_{\text{res}})] \leq C_1\mathbb{E}[\|J_s J_r^\top\|_F^2] + \delta_1. \tag{37}$$

**Step 2: Linking Jacobian overlap to Independence.** We relate the Jacobian cross-term to the feature independence. Consider the local linearization of the streams around the centered batch data $\tilde{H}$: $Z_s \approx J_s\tilde{H}$ and $Z_r \approx J_r\tilde{H}$. The empirical cross-covariance matrix $\Sigma_{sr}$ satisfies:

$$\Sigma_{sr} = \frac{1}{n}Z_s Z_r^\top \approx J_s\left(\frac{1}{n}\tilde{H}\tilde{H}^\top\right)J_r^\top \approx J_s J_r^\top, \tag{38}$$

assuming the backbone features are approximately whitened (e.g., via LayerNorm). Recall that CKA (Kornblith et al., 2019) approximates the squared Frobenius norm of the cross-covariance matrix. Thus, our independence measure implies:

$$\text{CKA}(Z_s, Z_r) \propto \|\Sigma_{sr}\|_F^2 \approx \|J_s J_r^\top\|_F^2. \tag{39}$$

Therefore, there exists a constant $C_2$ and residual $\delta_2$ (accounting for linearization error) such that:

$$\mathbb{E}[\|J_s J_r^\top\|_F^2] \leq C_2 \cdot \text{CKA}(Z_s, Z_r) + \delta_2. \tag{40}$$

**Step 3: Conclusion.** Substituting Equation 40 into Equation 37 and merging constants ($c = C_1 C_2$, $\delta = C_1\delta_2 + \delta_1$), we obtain the bound in Proposition 2.5:

$$\mathbb{E}[\cos^2(g_{\text{task}}, g_{\text{res}})] \leq c \cdot \text{CKA}(Z_s, Z_r) + \delta. \tag{41}$$

This confirms that minimizing the dependence between streams structurally suppresses the interference between task-relevant and nuisance gradients. $\square$

## E. Model architecture details

**Backbone and task heads (dense prediction).** On NYUv2, Cityscapes, GTA5, and Cityscapes-C, we use a dilated ResNet-50 encoder pretrained on ImageNet. For all dense tasks (segmentation, depth, normals), we use DeepLabV3+-style task heads. Given the semantic representation $Z_s$, we apply an ASPP module to capture multi-scale context, concatenate it with a low-level feature from the encoder (projected by a $1 \times 1$ convolution), and then apply $3 \times 3$ and $1 \times 1$ convolutions followed by bilinear upsampling to recover the input resolution. The segmentation head outputs $C$-channel logits, the depth head outputs a single-channel regression map, and the normal head outputs 3 channels followed by $\ell_2$ normalization. All baselines share the same backbone and task head architectures as our method.

**Backbone and heads on CelebA.** On CelebA, we also use a ResNet-50 backbone (with dilated convolutions disabled for classification) pretrained on ImageNet. After global average pooling, we apply a shared fully connected layer with 256 hidden units, BatchNorm, ReLU, and dropout (0.5), followed by a linear layer that outputs logits for 40 attributes. We train with the binary cross-entropy loss with logits.

**Semantic and residual projectors.** Our CORE-MTL model augments the encoder with two projectors: a semantic projector $P_s$ and a residual projector $P_r$. On dense prediction benchmarks, both $P_s$ and $P_r$ are implemented as $1 \times 1$ convolutions that map the combined encoder features (e.g., 2048 channels) to a lower-dimensional latent space (typically 256–512 channels). The semantic branch produces a spatial feature map $Z_s \in \mathbb{R}^{C_s \times H \times W}$ that preserves spatial resolution. The residual branch produces a feature map $P_r(H) \in \mathbb{R}^{C_r \times H \times W}$, which is then passed through global average pooling to yield a spatially invariant residual vector $z_r \in \mathbb{R}^{C_r}$. On CelebA, we use lightweight Conv–BN–ReLU blocks for $P_s$ and $P_r$.

**Decoder and counterfactual generator (training-time only).** The decoder $D$ takes $(Z_s, z_r)$ as input and reconstructs the input image (and optionally geometric signals) to support reconstruction and counterfactual augmentation losses. It is implemented as a stack of residual upsampling blocks without explicit skip connections from the encoder (i.e., not a full U-Net). For Causal Feature Augmentation (CFA), we generate counterfactual residual vectors by shuffling $z_r$ within a mini-batch, synthesize counterfactual images via $D(Z_s, z_r[\pi])$, and re-encode them through the encoder and projectors before feeding them to the task heads. The decoder and CFA pathway are used only at training time and are entirely removed at inference.

# F. Experimental Details

## F.1. Datasets and preprocessing

**NYUv2.** We follow the LibMTL and MTAN preprocessing. The dataset is split into 795 training and 654 validation images. We consider three tasks: 13-class semantic segmentation, depth estimation, and surface normal prediction. Inputs are resized to $288 \times 384$. For depth and normals, invalid pixels are masked out using a binary mask $\mathbf{1}[\sum_c \mathrm{GT}_c \neq 0]$ during both training and evaluation. Depth evaluation uses mean absolute error (Abs Err) and relative error (Rel Err) without additional log transforms or clipping. Normals are evaluated with mean and median angular error and the percentage of pixels within $11.25°$, $22.5°$, and $30°$.

**Cityscapes (ID).** For in-distribution multi-task learning, we use the Cityscapes subset and preprocessing provided by LibMTL. We adopt the standard 2975/500 train/val split and use two tasks: 7-class semantic segmentation and monocular depth estimation. The preprocessed images and labels are stored as $128 \times 256$ numpy arrays. Depth labels are obtained from the same preprocessed package (pseudo-depth derived from stereo disparity). We evaluate segmentation with mIoU and pixel accuracy, and depth with Abs Err and Rel Err.

## F.2. Optimization and hyperparameters

**Optimizer and learning-rate schedule.** Across all benchmarks (NYUv2, Cityscapes, GTA5→Cityscapes, Cityscapes-C, and CelebA), we use the AdamW optimizer with an initial learning rate of $2 \times 10^{-4}$ and weight decay $10^{-4}$. We adopt a cosine annealing learning-rate schedule with a minimum learning rate equal to $0.01$ times the initial value. A warmup phase is used at the beginning of training: 5 warmup epochs for NYUv2, Cityscapes, Cityscapes-C, and CelebA, and 10 warmup epochs for GTA5→Cityscapes.

**Batch sizes, epochs, and hardware.** We use the following batch sizes and training durations: NYUv2: batch size 16, 100 epochs; Cityscapes (ID): batch size 32, 100 epochs; GTA5→Cityscapes: batch size 32, 100 epochs; Cityscapes-C: batch size 16, 100 epochs (using the model trained on clean Cityscapes); CelebA: batch size 256, 50 epochs. All experiments are implemented in PyTorch and run on NVIDIA A800 GPUs (80GB); unless otherwise noted, each experiment uses a single GPU. These configurations are chosen such that each experiment fits comfortably within one GPU.

**Data augmentation.** On NYUv2, we apply random horizontal flipping, random scale-and-crop with resize scales in $\{1.0, 1.2, 1.5\}$ to a fixed input size, and color jitter with brightness, contrast, and saturation set to 0.4 and hue 0.1. On CelebA, we use random horizontal flipping and mild color jitter (brightness, contrast, and saturation 0.1, no hue change), followed by resizing to $128 \times 128$. For GTA5, Cityscapes, and Cityscapes-C, we follow the LibMTL preprocessing for

---

**Algorithm 1** CORE-MTL Training Procedure

---

 1: **Input:** dataset $\mathcal{D}$, batch size $B$, weights $\lambda_{\text{rec}}, \lambda_{\text{CKA}}, \lambda_{\text{CFA}}$, and $\lambda_{\text{lpips}}$, flag GROUNDINGTYPE (Hard/Soft)
 2: **Initialize:** encoder $E$, projectors $P_s, P_r$, task heads $\{f_t\}_{t=1}^K$, decoder $D$
 3: **repeat**
 4:    **1) Encode & Factorize**
 5:    Sample batch $\{(x_i, \{y_{t,i}\}_{t=1}^K)\}_{i=1}^B \sim \mathcal{D}$
 6:    $h_i \leftarrow E(x_i)$
 7:    $z_{s,i} \leftarrow P_s(h_i)$
 8:    $z_{r,i} \leftarrow \{P_r(h_i), \text{GAP}(h_i)\}$ {Spatial & global residual codes}
 9:    **2) Task Prediction (Semantics Only)**
10:    $\hat{y}_{t,i} \leftarrow f_t(z_{s,i})$
11:    $L_{\text{task}} \leftarrow \sum_{t=1}^K w_t \cdot \frac{1}{B} \sum_{i=1}^B \mathcal{L}_t(\hat{y}_{t,i}, y_{t,i})$
12:    **3) Grounding via Reconstruction**
13:    **if** HASPHYSICSPRIOR **then**
14:      $\hat{x}_i \leftarrow D_{\text{phy}}(z_{s,i}, z_{r,i})$ {Decodes Albedo & Lighting from $z_r$}
15:      $L_{\text{rec}} \leftarrow \frac{1}{B} \sum_{i=1}^B (\|x_i - \hat{x}_i\|_1 + \lambda_{\text{lpips}} \text{LPIPS}(x_i, \hat{x}_i))$
16:    **else**
17:      $\hat{x}_i \leftarrow D_{\text{gen}}(z_{s,i}, z_{r,i})$ {Generic convolutional decoding}
18:      $L_{\text{rec}} \leftarrow \frac{1}{B} \sum_{i=1}^B \|x_i - \hat{x}_i\|_1$
19:    **end if**
20:    **4) Structural Independence**
21:    $L_{\text{CKA}} \leftarrow \text{LinearCKA}(\{z_{s,i}\}_{i=1}^B, \{z_{r,i}\}_{i=1}^B)$ {Minimize semantic-residual dependence}
22:    **5) Counterfactual Feature Augmentation (CFA)**
23:    **if** GROUNDINGTYPE == Hard **then**
24:      $\pi \leftarrow \text{Shuffle}(1{:}B)$
25:      $x_i^{\text{CFA}} \leftarrow D_{\text{phy}}(z_{s,i}, z_{r,\pi(i)})$ {Synthesize via physical recombination}
26:      $h_i^{\text{CFA}} \leftarrow E(x_i^{\text{CFA}}); \; z_{s,i}^{\text{CFA}} \leftarrow P_s(h_i^{\text{CFA}})$
27:      $\hat{y}_{t,i}^{\text{CFA}} \leftarrow f_t(z_{s,i}^{\text{CFA}})$
28:      $L_{\text{CFA}} \leftarrow \sum_{t=1}^K w_t \cdot \frac{1}{B} \sum_{i=1}^B \mathcal{L}_t(\hat{y}_{t,i}^{\text{CFA}}, y_{t,i})$
29:    **else**
30:      $L_{\text{CFA}} \leftarrow 0$
31:    **end if**
32:    **6) Optimization**
33:    $L_{\text{total}} \leftarrow L_{\text{task}} + \lambda_{\text{rec}} L_{\text{rec}} + \lambda_{\text{CKA}} L_{\text{CKA}} + \lambda_{\text{CFA}} L_{\text{CFA}}$
34:    Update $\theta \leftarrow \theta - \eta \nabla_\theta L_{\text{total}}$
35: **until** convergence

---

resolution and normalization and apply random horizontal flipping as the primary augmentation. No additional photometric augmentation is used in these experiments.

### F.3. Training Algorithm

### F.4. Loss weights

Our total loss is

$$\mathcal{L}_{\text{total}} = \mathcal{L}_{\text{task}} + \lambda_{\text{rec}}\mathcal{L}_{\text{rec}} + \lambda_{\text{CKA}}\mathcal{L}_{\text{CKA}} + \lambda_{\text{CFA}}\mathcal{L}_{\text{CFA}},$$

where $\mathcal{L}_{\text{task}}$ aggregates the task-specific losses (segmentation, depth, normals or attribute classification), $\mathcal{L}_{\text{rec}}$ includes geometric and appearance reconstruction terms (and auxiliary $\ell_1$/perceptual terms when applicable), $\mathcal{L}_{\text{CKA}}$ is the independence regularizer between $Z_s$ and $Z_r$, and $\mathcal{L}_{\text{CFA}}$ is the consistency loss under counterfactual feature augmentation.

Table 7 summarizes the loss weights used in our experiments. For NYUv2, $\lambda_{\text{rec}}$ is reported as $(\alpha_{\text{geom}}, \beta_{\text{app}}, \lambda_{\text{L1}})$; for Cityscapes and GTA5→Cityscapes it is $(\alpha_{\text{geom}}, \beta_{\text{app}}, \lambda_{\text{img}})$.

*Table 7.* Loss weights used in our experiments. "Rec." lists $(\alpha_{\text{geom}}, \beta_{\text{app}}, \cdot)$ as described in the text.

| Dataset | $\lambda_{\text{seg}}$ | $\lambda_{\text{depth}}$ | $\lambda_{\text{normal}}$ | Rec. | $\lambda_{\text{CKA}}$ | $\lambda_{\text{CFA}}$ |
|---|---|---|---|---|---|---|
| NYUv2 | 20.0 | 10.0 | 10.0 | $(2.0, 15.0, 2.0)$ | 1.0 | 1.0 |
| Cityscapes (ID) | 10.0 | 20.0 | – | $(2.0, 15.0, 5.0)$ | 1.0 | 1.0 |
| GTA5→Cityscapes | 30.0 | 25.0 | – | $(5.0, 2.0, 5.0)$ | 0.1 | 0.5 |
| Cityscapes→Cityscapes-C | 10.0 | 20.0 | – | $(2.0, 15.0, 5.0)$ | 1.0 | 1.0 |
| CelebA | – | – | – | $(5.0, -, 1.0)$ | 0.5 | 0.0 |

### F.5. Hardware and software environment

All experiments are implemented in PyTorch and run on NVIDIA A800 GPUs with 80GB of memory. Although multiple GPUs are available, each experiment is conducted on a single GPU. This configuration is sufficient to train our model and all baselines with the batch sizes and resolutions described above.

## G. Sensitivity to Independence Regularization ($\lambda_{\text{CKA}}$)

In this section, we provide a detailed analysis of the trade-off between semantic independence and task performance by varying the independence loss weight $\lambda_{\text{CKA}}$.

**Training Dynamics.**  Figure 6 and Figure 7 illustrate the evolution of the independence metric (Linear CKA) during training on NYUv2 and the GTA5→Cityscapes transfer setting, respectively. As expected, higher $\lambda_{\text{CKA}}$ values lead to lower CKA scores, indicating stronger orthogonality between the semantic and residual streams.

**Impact on Performance.**  Table 8 and Table 9 present the downstream performance. On NYUv2, we observe that $\lambda_{\text{CKA}} = 1.0$ yields the best balance, improving segmentation and depth estimation over the unregularized baseline ($\lambda = 0$). However, setting $\lambda_{\text{CKA}}$ too high (100.0) degrades performance, particularly on surface normal prediction. This suggests that strictly orthogonalizing the representations may discard some useful shared information or over-constrain the semantic stream when the task difficulty is high.

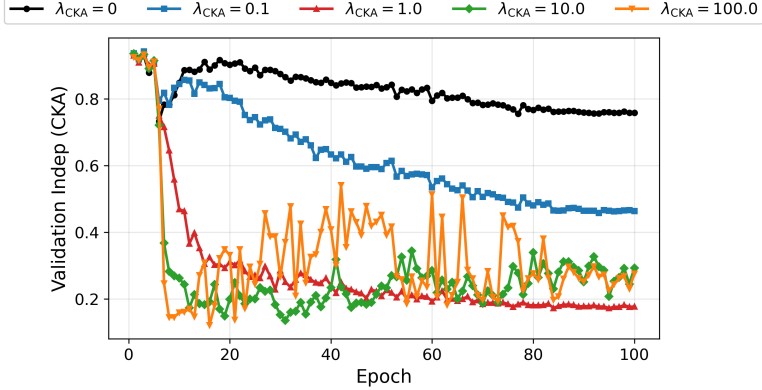

*Figure 6.* Independence(CKA) between $Z_s$ and $Z_r$ over epochs on NYUv2.

## H. Sensitivity to Counterfactual Augmentation ($\lambda_{\text{CFA}}$)

Table 10 demonstrates the effect of the counterfactual consistency loss weight $\lambda_{\text{CFA}}$ in the OOD setting (GTA5→Cityscapes). By enforcing the model to output consistent predictions when the residual style codes are swapped, CFA actively immunizes the semantic stream against spurious correlations.

We observe that a moderate weight (e.g., $0.5$ or $1.0$) significantly improves target domain performance (Tgt mIoU), indicating that the model successfully learns to be invariant to residual style factors. However, increasing $\lambda_{\text{CFA}}$ to $5.0$ leads to a

*Table 8.* Ablation on the CKA loss weight $\lambda_{\text{CKA}}$ on NYUv2.

| $\lambda_{\text{CKA}}$ | Seg | | Depth | | Normal | | | | |
|---|---|---|---|---|---|---|---|---|---|
| | mIoU↑ | Pix Acc↑ | Abs Err↓ | Rel Err↓ | Mean Ang↓ | Median↓ | Acc 11↑ | Acc 22↑ | Acc 30↑ |
| 0.0 | 0.5636 | 0.7719 | 0.3555 | 0.1433 | 22.9829 | 17.2257 | 0.3416 | 0.6090 | 0.7255 |
| 0.1 | 0.5676 | **0.7749** | 0.3570 | 0.1451 | 22.8554 | 17.0985 | 0.3441 | 0.6127 | 0.7291 |
| 1.0 | **0.5694** | 0.7735 | 0.3555 | 0.1445 | **22.7650** | **17.0039** | **0.3464** | **0.6141** | **0.7303** |
| 10.0 | 0.5647 | 0.7714 | **0.3509** | **0.1410** | 23.0706 | 17.4375 | 0.3390 | 0.6038 | 0.7220 |
| 100.0 | 0.5551 | 0.7656 | 0.3690 | 0.1484 | 23.9131 | 18.4297 | 0.3209 | 0.5821 | 0.7034 |

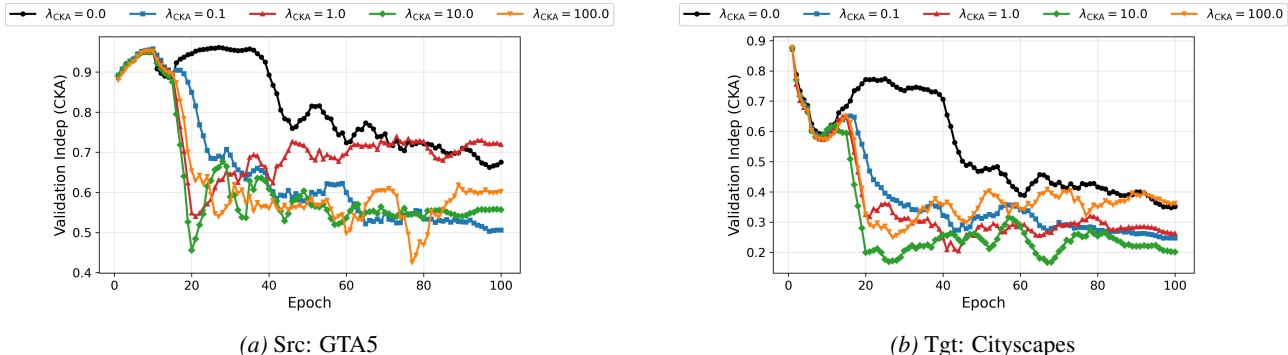

*(a)* Src: GTA5      *(b)* Tgt: Cityscapes

*Figure 7.* Validation Independence(CKA) between $Z_s$ and $Z_r$ on the source (GTA5) and target (Cityscapes) domains.

performance drop. This is likely because the perturbation noise becomes too strong relative to the task loss, effectively disrupting the learning of the primary semantic task on the source domain.

## I. Colored-Cityscapes Stress Test

**Motivation.** Assumption 2.2 uses an idealized latent SCM in which semantic factors and residual factors are separated for analysis. In real scenes, however, semantic content and contextual appearance are often strongly correlated. For example, road, sky, vegetation, and building regions usually co-occur with characteristic colors and textures. Such correlations can induce shortcut predictors that perform well in the source distribution but fail when the appearance statistics change. To examine whether CORE-MTL remains effective beyond this idealized setting, we construct a Colored-Cityscapes stress test that deliberately creates strong semantic–appearance dependence in the source distribution and breaks it in the target distribution.

**Dataset construction.** We build the stress test on top of the Cityscapes multi-task setting used in the main experiments. For the source distribution, each semantic class is assigned a fixed artificial color, and the input image is recolored according to its semantic mask. This creates a deterministic class–color association, making color a strong shortcut cue for semantic prediction. For the target distribution, we keep the same images, semantic labels, and depth labels, but permute the class–color mapping. Therefore, the target distribution preserves the underlying scene layout and task supervision while changing only the appearance shortcut. This controlled construction directly tests whether a method relies on residual appearance cues or learns a more stable task-relevant representation.

**Results.** As shown in Table 11, CORE-MTL achieves the best target-domain performance across all four metrics. In particular, under the permuted class–color mapping, CORE-MTL obtains the highest segmentation mIoU and pixel accuracy, as well as the lowest depth absolute and relative errors. These results show that CORE-MTL remains robust even when semantic content and appearance context are deliberately made strongly correlated in the source distribution, supporting that its semantic–residual factorization reduces residual-appearance leakage into the prediction stream.

*Table 9.* Ablation on the independence loss weight $\lambda_{\text{CKA}}$ on GTA5→Cityscapes. Src and Tgt denote GTA5 and Cityscapes, respectively.

| $\lambda_{\text{CKA}}$ | Seg mIoU | | Seg Pix Acc | | Depth Abs Err | | Depth Rel Err | |
|---|---|---|---|---|---|---|---|---|
| | Src↑ | Tgt↑ | Src↑ | Tgt↑ | Src↓ | Tgt↓ | Src↓ | Tgt↓ |
| 0.0 | 0.6540 | **0.5795** | 0.9439 | **0.8678** | 0.0187 | 0.1064 | **0.5161** | **252.3461** |
| 0.1 | **0.6583** | 0.5731 | 0.9439 | 0.8615 | 0.0181 | 0.1022 | 0.8236 | 252.6473 |
| 1.0 | 0.6449 | 0.5497 | **0.9445** | 0.8393 | 0.0222 | 0.1049 | 0.7117 | 252.9314 |
| 10.0 | 0.6573 | 0.5655 | 0.9438 | 0.8516 | **0.0169** | **0.0870** | 0.6980 | 252.8681 |
| 100.0 | 0.6550 | 0.5600 | 0.9429 | 0.8505 | **0.0169** | 0.1126 | 0.6595 | 290.2770 |

*Table 10.* Ablation on the counterfactual augmentation weight $\lambda_{\text{CFA}}$ on GTA5→Cityscapes. Src and Tgt denote GTA5 and Cityscapes, respectively.

| $\lambda_{\text{CFA}}$ | Seg mIoU | | Seg Pix Acc | | Depth Abs Err | | Depth Rel Err | |
|---|---|---|---|---|---|---|---|---|
| | Src↑ | Tgt↑ | Src↑ | Tgt↑ | Src↓ | Tgt↓ | Src↓ | Tgt↓ |
| 0 | 0.6605 | 0.5303 | **0.9471** | 0.8267 | **0.0170** | 0.1207 | **0.3646** | 272.0598 |
| 0.1 | 0.6615 | 0.5174 | 0.9452 | 0.8399 | 0.0186 | **0.0924** | 0.4699 | **242.8646** |
| 0.5 | 0.6466 | **0.5657** | 0.9393 | **0.8558** | 0.0171 | 0.1028 | 0.6887 | 255.1533 |
| 1.0 | 0.6537 | 0.5572 | 0.9440 | 0.8532 | 0.0179 | 0.1065 | 0.7325 | 263.7881 |
| 5.0 | **0.6655** | 0.5206 | 0.9460 | 0.8272 | 0.0181 | 0.1128 | 0.9077 | 265.3829 |

## J. Cityscapes→Cityscapes-C per-corruption results

In this section, we provide a detailed breakdown of the performance on individual corruption types from the Cityscapes-C benchmark. While the main text reports the averaged robustness metrics (see Table 2), Tables 12 through 15 present the specific results for Contrast, Defocus Blur, Fog, and Gaussian Noise corruptions, respectively.

Consistent with the aggregated results, **CORE-MTL** demonstrates superior robustness across these diverse domain shifts. Notably, in scenarios characterized by severe visibility reduction, such as Fog (Table 14), our method significantly outperforms gradient-balancing baselines (e.g., surpassing GradNorm and PCGrad by a large margin in mIoU). Similarly, under high-frequency noise interference like Gaussian Noise (Table 15), our approach maintains higher segmentation accuracy and substantially lower depth relative error. These consistent improvements reinforce our claim that structurally disentangling the invariant semantic stream ($Z_s$) effectively isolates task predictions from residual variations ($Z_r$).

## K. CelebA attribute list (ordered)

In this section, we provide the complete list of 40 facial attributes from the CelebA dataset, presented in the specific order used across our scalability and gradient orthogonality experiments.

Table 16 details the mapping between the attribute names and their corresponding indices. This ordering is crucial for interpreting the gradient interaction heatmaps presented in Figure 5 of the main text. By referencing this table, readers can correlate the high-conflict or orthogonal regions in the heatmaps with specific semantic attribute pairs (e.g., correlating index 21 "Male" with index 37 "Wearing Lipstick" to understand specific task interferences), thereby providing a granular view of how CORE-MTL mitigates optimization conflicts compared to baseline methods.

*Table 11.* Colored-Cityscapes stress test. Class colors are fixed in the source distribution and permuted in the target distribution. Higher is better for mIoU and pixel accuracy, while lower is better for depth errors.

| Method | Source mIoU↑ | Source Pix Acc↑ | Source Abs Err↓ | Source Rel Err↓ | Target mIoU↑ | Target Pix Acc↑ | Target Abs Err↓ | Target Rel Err↓ |
|---|---|---|---|---|---|---|---|---|
| EW | 0.7811 | 0.9400 | 0.0123 | 42.7581 | 0.3356 | 0.4501 | 0.0319 | 47.0544 |
| PCGrad | 0.7800 | 0.9396 | 0.0125 | 42.1546 | 0.3525 | 0.4282 | 0.0370 | 46.6852 |
| GradNorm | 0.7795 | 0.9398 | 0.0121 | 43.0793 | 0.3545 | 0.4284 | 0.0409 | 46.6124 |
| STCH | 0.7719 | 0.9375 | **0.0119** | 42.8746 | 0.3548 | 0.4457 | 0.0251 | 44.4525 |
| MTAN | 0.7902 | 0.9421 | 0.0121 | 42.8665 | 0.3597 | 0.4204 | 0.0513 | 46.1531 |
| CORE-MTL (Ours) | **0.8295** | **0.9492** | 0.0121 | **28.2662** | **0.3738** | **0.4889** | **0.0249** | **32.1862** |

*Table 12.* Cityscapes→Cityscapes-C: Contrast corruption.

| Method | Seg mIoU↑ | Seg Pix Acc↑ | Depth Abs Err↓ | Depth Rel Err↓ |
|---|---|---|---|---|
| Equal Weighting | 0.5300 | 0.8354 | 0.0267 | 74.2265 |
| PCGrad | 0.5282 | 0.8286 | 0.0281 | 75.4339 |
| GradNorm | 0.5364 | 0.8331 | 0.0240 | 70.8020 |
| STCH | 0.5398 | 0.8372 | 0.0224 | 65.9642 |
| MTAN | 0.5377 | 0.8337 | 0.0242 | 66.0131 |
| MOML | **0.5568** | **0.8482** | 0.0234 | 61.1203 |
| RLW | 0.5396 | 0.8370 | 0.0241 | 73.1554 |
| ExcessMTL | 0.5140 | 0.8261 | 0.0280 | 76.8556 |
| FairGrad | 0.5271 | 0.8288 | **0.0206** | 62.1508 |
| RepMTL | 0.5434 | 0.8389 | 0.0208 | 60.3663 |
| **CORE-MTL (Ours)** | 0.5405 | 0.8325 | 0.0234 | **46.5571** |

*Table 13.* Cityscapes→Cityscapes-C: Defocus Blur corruption.

| Method | Seg mIoU↑ | Seg Pix Acc↑ | Depth Abs Err↓ | Depth Rel Err↓ |
|---|---|---|---|---|
| Equal Weighting | 0.6876 | 0.9162 | 0.0133 | 44.4824 |
| PCGrad | 0.6903 | 0.9159 | 0.0133 | 45.3982 |
| GradNorm | 0.6934 | 0.9168 | 0.0129 | 45.4517 |
| STCH | 0.6877 | 0.9151 | 0.0124 | 43.4140 |
| MTAN | 0.6924 | 0.9169 | 0.0132 | 46.7692 |
| MOML | 0.6923 | 0.9166 | 0.0143 | 45.8430 |
| RLW | 0.6913 | 0.9156 | 0.0134 | 45.0699 |
| ExcessMTL | 0.6879 | 0.9164 | 0.0133 | 44.5311 |
| FairGrad | 0.6892 | 0.9159 | **0.0123** | 43.8715 |
| RepMTL | **0.7110** | **0.9202** | **0.0123** | 39.4019 |
| **CORE-MTL (Ours)** | 0.7079 | 0.9191 | 0.0128 | **28.8479** |

*Table 14.* Cityscapes→Cityscapes-C: Fog corruption.

| Method | Seg mIoU↑ | Seg Pix Acc↑ | Depth Abs Err↓ | Depth Rel Err↓ |
|---|---|---|---|---|
| Equal Weighting | 0.4567 | 0.7580 | 0.0399 | 66.4706 |
| PCGrad | 0.4667 | 0.7721 | 0.0358 | 65.9890 |
| GradNorm | 0.4696 | 0.7630 | 0.0334 | 69.7784 |
| STCH | 0.4556 | 0.7654 | 0.0280 | 63.6113 |
| MTAN | 0.4574 | 0.7673 | 0.0353 | 65.5247 |
| MOML | 0.4893 | 0.7934 | 0.0422 | 58.9439 |
| RLW | 0.4826 | 0.7881 | 0.0312 | 59.9741 |
| ExcessMTL | 0.4625 | 0.7705 | 0.0364 | 71.2259 |
| FairGrad | 0.4459 | 0.7531 | 0.0286 | 64.3044 |
| RepMTL | 0.4391 | 0.7670 | 0.0247 | 64.6777 |
| **CORE-MTL (Ours)** | **0.5165** | **0.8143** | **0.0219** | **43.1408** |

*Table 15.* Cityscapes→Cityscapes-C: Gaussian Noise corruption.

| Method | Seg mIoU↑ | Seg Pix Acc↑ | Depth Abs Err↓ | Depth Rel Err↓ |
|---|---|---|---|---|
| Equal Weighting | 0.6580 | 0.8988 | 0.0156 | 49.8246 |
| PCGrad | 0.6553 | 0.8915 | 0.0150 | 49.9915 |
| GradNorm | 0.6607 | 0.8961 | 0.0148 | 50.1081 |
| STCH | 0.6580 | 0.8971 | **0.0138** | 45.2761 |
| MTAN | 0.6674 | 0.9017 | 0.0146 | 47.8989 |
| MOML | 0.6572 | 0.8951 | 0.0158 | 50.1173 |
| RLW | 0.6528 | 0.8919 | 0.0160 | 49.0778 |
| ExcessMTL | 0.6610 | 0.9004 | 0.0153 | 48.5149 |
| FairGrad | 0.6601 | 0.8980 | **0.0138** | 47.5155 |
| RepMTL | 0.6727 | 0.8978 | 0.0144 | 41.5611 |
| **CORE-MTL (Ours)** | **0.6766** | **0.9023** | 0.0146 | **33.8447** |

*Table 16.* CelebA attributes in the official order.

| | | | |
|---|---|---|---|
| 1 5_o_Clock_Shadow | 11 Blurry | 21 Male | 31 Sideburns |
| 2 Arched_Eyebrows | 12 Brown_Hair | 22 Mouth_Slightly_Open | 32 Smiling |
| 3 Attractive | 13 Bushy_Eyebrows | 23 Mustache | 33 Straight_Hair |
| 4 Bags_Under_Eyes | 14 Chubby | 24 Narrow_Eyes | 34 Wavy_Hair |
| 5 Bald | 15 Double_Chin | 25 No_Beard | 35 Wearing_Earrings |
| 6 Bangs | 16 Eyeglasses | 26 Oval_Face | 36 Wearing_Hat |
| 7 Big_Lips | 17 Goatee | 27 Pale_Skin | 37 Wearing_Lipstick |
| 8 Big_Nose | 18 Gray_Hair | 28 Pointy_Nose | 38 Wearing_Necklace |
| 9 Black_Hair | 19 Heavy_Makeup | 29 Receding_Hairline | 39 Wearing_Necktie |
| 10 Blond_Hair | 20 High_Cheekbones | 30 Rosy_Cheeks | 40 Young |

