# OpenReview forum: "CORE-MTL: Rethinking Gradient Balancing via Causal Orthogonal Representations"
_ICML.cc/2026/Conference — ICML 2026 regular_

### Official Review · Reviewer_Bgp8 · 2026-02-23

**Soundness:** 3
**Presentation:** 3
**Significance:** 3
**Originality:** 3
**Overall Recommendation:** 4
**Confidence:** 5

**Summary:**

This paper aims to address the issues of negative transfer and poor out-of-distribution generalization in multi-task learning caused by shared representation entanglement. The authors point out that most existing optimization-centric MTL methods merely manipulate gradients while neglecting the causal structure of the feature space. To this end, the authors propose the CORE-MTL framework, which structurally disentangles the shared representation into a task-relevant semantic stream ($Z_s$) and a task-irrelevant residual stream ($Z_r$) to alleviate gradient conflicts at their root. This framework incorporates a structural independence constraint, physical/generic reconstruction priors, and a counterfactual augmentation  mechanism. Theoretical and empirical results demonstrate that CORE-MTL achieves improvements in both in-distribution and out-of-distribution benchmarks, and it claims to intrinsically achieve gradient orthogonality.

**Compliance With Llm Reviewing Policy:**

Affirmed.

**Final Justification:**

The authors answered my questions to the point and admitted some shortcomings in their article. Given the existence of a number of limitations, I ultimately prefer this to be a border article.

**Key Questions For Authors:**

1. In the early stages of training, the disentanglement into $Z_s$ and $Z_r$ is bound to be inaccurate. Executing the proposed CounterFactual Path at this stage could easily lead to training instability. Does this phenomenon occur during training, and what specific training strategies were employed to handle it?
2. In the "GTA5 $\rightarrow$ Cityscapes" experiment in Table 2, why does the Rel Err metric on the source domain perform significantly worse than the baselines? This looks like anomalous data. How to explain this?
3. Are the labels for $Z_s$ and $Z_r$ in Figure 3 swapped? The visualized distributions look somewhat problematic. How to understand the Figure?
4. The authors claim in the text that CORE-MTL "achieves near-zero cosine similarity" and that the heatmaps exhibit a "structured, block-diagonal alignment," using this to prove the mitigation of negative transfer. However, looking at the actual Figure 5, panel (b) for PCGrad is almost entirely white, whereas panel (d) for CORE-MTL is filled with red and blue blocks. Visually, this suggests that the interactions and dependencies between gradients have been intensified rather than diminished. How to understand it?

**Limitations:**

The authors did not discuss the limitations of their work. I believe the limitations primarily lie in two aspects:
1. The empirical evaluation section needs to be further expanded and enriched to increase persuasiveness.
2. The conceptualization of the method requires more scientific rigor. The current approach leans towards the idealized side, leaving significant room for improvement regarding its practical applicability.

**Strengths And Weaknesses:**

## Strength
1. The motivation of the paper is highly inspiring. It attributes the common gradient conflict issue in MTL to a deeper problem of "shared feature entanglement," shifting the paradigm from the traditional "optimization-centric" approach to a "representation-centric" disentanglement perspective.
2. The authors provide a clear theoretical framework proving that under feature entanglement, pure gradient balancing methods have an unavoidable OOD error lower bound. Concurrently, they establish a generalization error upper bound for the disentangled architecture. Overall, the theoretical derivations and algorithm design echo each other well.

## Weakness
1. Regarding Assumption 2.2, there is often a strong statistical dependency between background information and semantic information in real-world scenarios. Achieving strict independence between the two is highly challenging.
2. In Assumption 2.2, the features $Z_s$ and $Z_r$ are defined as being mixed via a rotation matrix $R_\psi$. However, in Section 3, the proposed framework's operation to disentangle features clearly misaligns with the definition in Section 2. The framework's setup acts more like feature selection----selecting a subset of features from the complete feature $Z$ as $Z_s$ and another subset as $Z_r$. This fundamentally contradicts the rotation-based mixing mechanism introduced in Section 2.
3. The baselines selected for comparison are somewhat outdated, making the empirical performance gains less convincing.
4. The authors introduce a considerable number of hyperparameters. Fine-tuning these hyperparameters could incur significant time and computational costs.
5. The authors anchor all downstream tasks to the same $Z_s$. However, the definitions of "semantics" (signal) versus "residuals" (noise) are inherently task-dependent. For instance, texture information might be noise for surface normal estimation but a critical cue for distinguishing specific categories in semantic segmentation. Applying a one-size-fits-all reconstruction prior to strip away so-called "irrelevant" features can easily fail when task heterogeneity is high. The authors' assumption here appears somewhat overly idealized.

I would be very willing to raise my score if the authors can satisfactorily address all of my concerns.

---

> ### Author Rebuttal · Authors · 2026-03-29
>
> We feel truly fortunate that our paper received such an exceptionally careful and insightful reading from a reviewer who engaged so deeply with our work!
>
> **W1**: We agree that semantic content and context can be strongly correlated in real scenes. To test whether CORE-MTL remains effective under such dependence, we construct a colored-Cityscapes setting where each class has a fixed color in training/ID (e.g., road is always blue) and the class-color mapping is permuted in OOD (e.g., road becomes red).
> |Method|ID mIoU|ID PixAcc|ID Abs|ID Rel|OOD mIoU|OOD PixAcc|OOD Abs|OOD Rel|
> |-|-|-|-|-|-|-|-|-|
> |STCH|0.7719|0.9375|**0.0119**|42.8746|0.3548|0.4457|0.0251|44.4525|
> |MTAN|0.7902|0.9421|0.0121|42.8665|0.3597|0.4204|0.0513|46.1531|
> |**Ours**|**0.8295**|**0.9492**|0.0121|**28.2662**|**0.3738**|**0.4889**|**0.0249**|**32.1862**|
>
> **W2**: We respectfully clarify that Assumption 2.2 describes the entanglement process inherent in standard encoders. Our projectors $P_s$ and $P_r$ (Section 3.1) are not simple feature selectors or slicers; instead, they learn a non-linear **unmixing transformation** that serves as the practical counterpart of inverting the entanglement rotation $R_\psi$. By mapping the shared representation into a more orthogonalized space $(Z_s,Z_r)$, the model structurally addresses the mixing identified in our theory.
>
> **W3**: We include three recent 2024–2025 baselines, and CORE-MTL still achieves the strongest overall performance on NYUv2.
> |Method|mIoU|PixAcc|AbsErr|RelErr|Mean|Med|11.25|22.5|30|
> |-|-|-|-|-|-|-|-|-|-|
> |ExcessMTL (ICML 2024)|0.4846|0.7180|0.3877|0.1624|26.1446|19.8459|0.3043|0.5499|0.6676|
> |FairGrad (ICML 2024)|0.5291|0.7427|0.3944|0.1585|23.0717|**16.3569**|**0.3657**|**0.6239**|0.7312|
> |RepMTL (ICCV 2025)|0.5492|0.7598|0.3727|0.1503|24.5348|19.6139|0.3018|0.5576|0.6847|
> |**Ours**|**0.5693**|**0.7759**|**0.3544**|**0.1466**|**22.4927**|16.8337|0.3501|0.6196|**0.7360**|
>
> **W4**: We agree that CORE-MTL introduces auxiliary coefficients. However, **Tables 7–9 show that performance mainly drops only at extreme values**, rather than being broadly brittle to tuning.
>
> **W5**: We do **not** claim that a universal “one-size-fits-all” $z_s$ exists for arbitrary heterogeneous tasks. For the dense scene understanding tasks studied here, a shared structural core is a reasonable inductive bias. Crucially, appearance and texture cues are **preserved, not discarded**: they are encoded in the residual stream ($z_r$). We agree that scaling to more heterogeneous task sets may require moving beyond a single shared $z_s$, which we will explicitly discuss as a limitation.
>
> **Q1**: This question pinpoints a training difficulty that only a very careful and experienced reviewer would notice. To prevent instability, we use a staged schedule: Stage 0 warms up the decomposition, Stage 1 adapts task heads with the counterfactual/residual path still off, and Stage 2 enables the full path. Moreover, the independence weight is kept at 0 before Stage 2 and warmed up only afterwards to avoid early feature collapse.
>
> **Q2:** Thank you for catching this anomaly. After re-checking the evaluation, we found an **implementation mismatch in source-domain Rel Err** only in gta5: the **official LibMTL depth metric uses a valid-depth mask (gt > 1e-6)**, while our evaluation omits it (**verifiable in the official implementation**). Since Rel Err divides by depth, this mainly inflates the **smallest-depth regime**. With the official protocol, the **source Rel Err becomes 0.1882**. We will correct this in the revision.
>
> **Q3**: Thanks for catching this. **Yes, Fig. 3(b) was mislabeled** (Fig. 3(a) is correct), and we will fix it in the revised version. The correct meaning is that **$z_s$ shifts less, while $z_r$ shifts more**, consistent with Table 3.
>
> **Q4**: Fig. 4 and Fig. 5 measure different quantities: **Fig. 4 is task-vs-reconstruction**, whereas **Fig. 5 is task-vs-task**. Thus, the **near-zero cosine similarity** statement refers to **Fig. 4**, not Fig. 5. Fig. 5 is therefore **not expected to be uniformly white**; its block pattern indicates that gradient relations become **more structured/modularized**, rather than simply vanishing. We will clarify it in the revision.
>
> **L1**: We agree that broader empirical coverage can further increase persuasiveness. This is why we add new recent baselines and additional shortcut-based colored-Cityscapes experiment in this rebuttal. We believe these additions substantially strengthen the empirical support of the paper.
>
> **L2**: We agree that the current formulation is partly idealized, and we will make these limitations explicit in the revision. 1) Our theory relies on a stylized linear-Gaussian assumption, and extending these guarantees to deep non-linear representations remains future work. 2) Although CORE-MTL works well on the dense scene understanding tasks studied here, scaling it to more heterogeneous or multimodal task sets may require more flexible grounding mechanisms.

---

> > ### Author Rebuttal · Reviewer_Bgp8 · 2026-04-01
> >
> > The authors have answered some questions and actually admitted that some opinions are real shortcomings in their work. I will also refer to the opinions of other reviewers and consider whether to adjust my score.

---

> > > ### Author Response · Authors · 2026-04-07
> > >
> > > Thank you again for the exceptionally careful reading and for considering our rebuttal so thoughtfully. We sincerely appreciate the rigor of your evaluation and fully respect your judgment.
> > >
> > > We are encouraged that the paper’s core motivation resonated: it reframes gradient conflict in MTL as a deeper shared-feature entanglement problem, shifting the perspective from optimization-centric balancing to representation-centric factorization, and it supports this view with a theory that explains the limitation of optimization-only methods under entanglement while motivating a disentangled architecture.
> > >
> > > In the rebuttal, we directly strengthened the paper on the most actionable concerns by adding recent 2024–2025 baselines, introducing a shortcut-based Colored-Cityscapes stress test, clarifying the staged training strategy for stability, and identifying the source-domain Rel Err anomaly as an evaluation mismatch rather than a conceptual failure. We hope these additions materially improve both the empirical support and the technical clarity of the work. More broadly, Section 2 should be read as a theoretical analysis tool that makes our representation-centric diagnosis mathematically explicit, rather than as a literal description of real-world data. Under this reading, the assumptions delimit the scope of the analysis, but do not undermine the paper’s central claim: shared feature entanglement is a root cause of negative transfer, and addressing it at the representation level is both theoretically motivated and empirically effective.
> > >
> > > This is also why we added Colored-Cityscapes in the rebuttal. It is a deliberately constructed stress test with strong semantic–context correlation through shortcut cues. The fact that CORE-MTL remains effective in this setting shows that our main idea does not depend on real scenes satisfying strict independence literally; instead, it supports the broader point that reducing representational entanglement improves robustness even under strong correlation. We hope these clarifications and added results are helpful as you finalize your assessment.
> > >
> > > We greatly appreciate your careful evaluation, your recognition of the paper’s core strengths, and your constructive suggestions, all of which will help us improve the final version of the paper.

---

### Official Review · Reviewer_JxLr · 2026-03-05

**Soundness:** 2
**Presentation:** 3
**Significance:** 3
**Originality:** 3
**Overall Recommendation:** 4
**Confidence:** 4

**Summary:**

The authors reinterpret gradient conflict not as the root cause of negative transfer, but rather as a symptom arising from entangled shared representations.
Based on this perspective, CORE-MTL decomposes the shared representation into a semantic stream and a residual stream.
Task heads are restricted to accessing only the semantic stream, while the framework integrates several mechanisms: CKA-based independence regularization, reconstruction-based grounding, and counterfactual feature augmentation (CFA) under a hard-grounded setting.
For structured scenes, the model employs a physics-based decoder, whereas in settings where explicit physical equations are difficult to specify, a generic convolutional decoder is used. This framing itself is conceptually interesting.
In particular, the claim that 'gradient balancing alone cannot resolve representation entanglement directly' challenges a well-established narrative in the multi-task learning literature.

**Compliance With Llm Reviewing Policy:**

Affirmed.

**Final Justification:**

Most of my concerns have been addressed. I will maintain my score.

**Key Questions For Authors:**

1. Is it possible to show  connection to causal variable identification and do-intervention justification empirically or theoretically?
2. Are there any evidences that show CFA is not just representation regularization and it is real causal intervention using counterfactual augmentation?
3. Are there more explanations about the relation between identifiability and soft-grounding? Why is it possible to claim identifiabillity on soft-groudning?

**Limitations:**

The work appears substantially stronger when viewed as a disentanglement or robust representation learning study rather than as a causal paper. Moderating the causal framing would likely clarify and better highlight the paper’s actual contributions. However, if these issues are adequately addressed, the paper could become substantially stronger while retaining its current causal framing.

**Strengths And Weaknesses:**

Strengths
1. The problem formulation is intellectually well-motivated. By reframing negative transfer as an issue of representation geometry rather than solely gradient geometry, the paper provides a plausible conceptual lens. This perspective is particularly helpful in explaining why post-hoc gradient surgery approaches frequently reach intrinsic limitations.
2. Strong empirical signal and scalability story.
3. Some analysis is implicitly coherent.
---
Weaknesses
1. While the contributions are potentially interesting as a representation learning approach, the paper presents them as if they constitute a causal identification result. The theorem 2.3 and 2.4 is relied on the detailed hypothesis and Appendix B provides useful surrogate but it does not have enough connection to causal variable identification and do-intervention justification.
2. It seems that there is exaggeration on causal intervention which is just representation regularization. The paper shuffles the learned residual code and generates the synthetic images with consistency loss. This is just model-internal recombination so it seems that it is different from explicit interventions for observable nuisance variable.

---

> ### Author Rebuttal · Authors · 2026-03-29
>
> We greatly appreciate your clear distinction between **causal identification** and a **causally motivated representation-learning framework**. This observation is particularly helpful in refining our framing and clarifying the level of causal interpretation that our results support.
>
> **1. Causal Framing & CFA (addressing W1, W2, Q1, Q2, L1).** We agree that claiming formal **causal identification** or explicit $do$-interventions on observable real-world variables would overstate our contribution. We will revise the manuscript to position CORE-MTL more precisely as a **causally motivated representation-learning framework**.
>
> **Theory.** Together, Theorems 2.3 and 2.4 should be interpreted as **structural generalization results** under the assumed latent SCM: Theorem 2.3 shows that entangled representations induce an irreducible OOD error floor, while Theorem 2.4 shows that reducing residual leakage tightens the target-domain robustness bound. This is why semantic--nuisance separation is central in our framework.
>
> **CFA.** We agree that our current evidence is **not sufficient to claim** that CFA is a real $do$-intervention on observable nuisance variables. What we can support is a narrower point: CFA is **not an arbitrary regularization term**, because it keeps $Z_s$ fixed and changes only $Z_r$. Empirically, **Appendix Table 9** shows that CFA is useful and non-redundant, while **Figure 2** and **Figure 3 / Table 3** show that this operation mainly changes residual-related variation while keeping semantics relatively stable. We will revise the manuscript to make this distinction explicit and avoid overstating CFA as full causal intervention.
>
> **2. Soft Grounding & Identifiability (addressing Q3, W1, L1).** We agree that soft grounding can **not** support a claim of **strict identifiability** without strong physical priors, and we will make this limitation explicit. While hard grounding provides stronger semantic anchoring through a physics-based decoder, soft grounding is better interpreted as **functional disentanglement**. By restricting task heads to $Z_s$, enforcing statistical independence via $L_{CKA}$, and requiring reconstruction from complementary factors, soft grounding encourages task-stable information to concentrate in $Z_s$ and nuisance variation in $Z_r$. Our **hard-vs.-soft grounding comparison** on **NYUv2** is consistent with this **functional-disentanglement interpretation**:
>
> |Setting|mIoU|Pixel Acc.|Depth Abs.|Depth Rel.|Mean (°)|Median (°)|Acc 11°|Acc 22°|Acc 30°|
> |---|---|---|---|---|---|---|---|---|---|
> |Soft Grounding|0.5652|0.7715|0.3555|**0.1427**|23.0044|17.3100|0.3415|0.6068|0.7238|
> |Hard Grounding|**0.5693**|**0.7759**|**0.3544**|0.1466|**22.4927**|**16.8337**|**0.3501**|**0.6196**|**0.7360**|
>
> This shows that **hard grounding is stronger overall**, while **soft grounding remains competitive** and even slightly improves Depth Rel, indicating that this weaker functional separation can still be useful even without strict identifiability.
>
> **3.Main Contribution.** Crucially, this functional separation also resolves interference at its geometric source. As Proposition 2.5 suggests, disentanglement induces **gradient orthogonality** as a structural by-product. In this sense, moderating the causal narrative strengthens, rather than weakens, our central technical message: restructuring the latent space can mitigate task interference **at the representation level rather than only through post-hoc gradient surgery**.

---

> > ### Author Rebuttal · Reviewer_JxLr · 2026-04-02
> >
> > 1. After your rebuttal, what exact causal claim still remains in the paper?
> > 2. If CFA is not a true intervention on observable nuisances, what evidence shows it is more than latent regularization or augmentation?
> > 3. Under soft grounding, what identifiability claim, if any, still remains, and how is it different from hard grounding?
> > 4. What evidence shows the gains come from structural disentanglement itself, rather than decoder capacity or regularization side effects?

---

> > > ### Author Response · Authors · 2026-04-05
> > >
> > > We thank the reviewer for these precise follow-up questions. The causal content of the paper is **not** a claim of causal discovery, identification of observable variables, or literal do-intervention on real-world nuisances; it is a **representation-level claim** that, under an **assumed latent Structural Causal Model (SCM)**, it identifies semantic–nuisance entanglement as a fundamental source of negative transfer, and shows that reducing such entanglement improves OOD robustness and mitigates task interference **beyond what post-hoc gradient surgery alone can guarantee**. We will revise our paper accordingly.
> > >
> > > **Q1:** The remaining causal claims in our paper are strictly conditional on our assumed **latent Structural Causal Model (SCM)**. Through this causal lens, the exact causal claims that remain are the following:
> > >
> > > (1) **A causal claim on structural limits under the assumed SCM (Theorem 2.3)**. Under our assumed SCM, entangling stable semantic factors with environment-dependent nuisances creates an irreducible OOD error floor. If invariant and spurious factors remain mixed, optimization alone cannot fully recover robustness.
> > >
> > > (2) **A causal claim on factor-role separation (Theorem 2.4).** Predictions should rely on stable semantics, not nuisance variations. Architectures restricting prediction to a semantic stream achieve tighter OOD bounds. This supports a causally inspired architectural bias rather than claiming discovery of true causal variables.
> > >
> > > (3) **A causal claim on task interference as a downstream symptom (Proposition 2.5).** Gradient conflict is a downstream consequence of entangling stable semantics with nuisances. By enforcing semantic–residual separation, our method directly reduces this representation-level interference.
> > >
> > > In summary, our core claim is a causally motivated structural one: aligning prediction with stable semantics and decoupling it from environmental nuisances improves robustness and reduces task interference. This highlights a **causal inductive bias** in MTL representations, **not strict causal identifiability**.
> > >
> > > **Q2:**  CFA is **not** a Pearlian intervention on real-world observables, **nor** a blind latent regularizer, but a **structurally grounded, representation-level operator implemented as a training objective**. A blind latent regularizer penalizes representations **globally**; CFA performs a **role-specific** counterfactual substitution and then tests invariance under that substitution. This substitution is role-specific because the physics-based decoder $D_{\text{phy}}$ imposes an explicit image-formation prior, so the operation approximates a controlled change in residual appearance while keeping the underlying semantic structure fixed, rather than applying an undirected perturbation in latent space.
> > >
> > > **Q3:**  Under soft grounding, we do **not** retain an identifiability claim; what remains is only a **functional-disentanglement claim**. Soft grounding encourages a useful partition where $\hat Z_s$ captures information for downstream prediction, while $\hat Z_r$ absorbs complementary variation for reconstruction. However, it does not guarantee unique recovery of the true underlying factors, which is consistent with existing impossibility results that **weak structural constraints are generally insufficient to guarantee unique recovery of causal latent factors from raw observational data[1][2]**.
> > >
> > > The difference lies in the **physical grounding strength** of **structural anchoring**. Hard grounding imposes an **explicit physics-based** image-formation model, which ties $Z_s$ more directly to geometry-related content and $Z_r$ more directly to photometric variation. Conversely, soft grounding's generic reconstruction objective lacks this physical bottleneck.
> > >
> > > **Q4:** The reduction in **inter-task gradient conflict in the shared backbone** is difficult to explain by decoder capacity or generic regularization alone: increasing decoder capacity only enlarges the set of realizable solutions, and generic regularization typically encourages smoothness or robustness, but **neither mechanism explicitly enforces the directional constraint** that task-relevant semantics should concentrate in $Z_s$ while nuisance variation should be absorbed by $Z_r$. Our method imposes exactly this structural split, so the observed conflict reduction is more naturally explained by **structural disentanglement itself**. In MTL, this is a genuine gain because shared parameters must serve multiple tasks simultaneously; when their gradients conflict, improving one task can directly harm another, i.e., negative transfer[3].
> > >
> > > [1] Locatello F, Bauer S, Lucic M, et al. Challenging common assumptions in the unsupervised learning of disentangled representations. ICML 2019.
> > >
> > > [2] Brehmer J, De Haan P, Lippe P, et al. Weakly supervised causal representation learning. NeurIPS 2022.
> > >
> > > [3] Liu B, Liu X, Jin X, et al. Conflict-averse gradient descent for multi-task learning. NeurIPS 2021.

---

### Official Review · Reviewer_mVbL · 2026-03-11

**Soundness:** 3
**Presentation:** 3
**Significance:** 3
**Originality:** 3
**Overall Recommendation:** 4
**Confidence:** 2

**Summary:**

This paper proposes CORE-MTL, a representation-centric framework that structurally disentangles the shared representation into semantic and residual streams, concentrating task-relevant structure in the semantic stream while relegating nuisance variation to the residual stream. The CORE-MTL method's performance is good. The motivation is clear and the method seems reasonable.

**Compliance With Llm Reviewing Policy:**

Affirmed.

**Final Justification:**

I will keep my score.

**Key Questions For Authors:**

1. Need ablation studies on all additional losses.

2. I have questions on the Depth result on Table 1. It seems all baselines can only achieve about Rel Err 37. But the proposed method achieves 19.66. The improvement is impressive. Is it true? Why in another metric on Depth (Abs Err), the proposed method achieve similar performance with baselines? Is the code correct?

3. MTAN is a different model. STCH is a weighting method. I wonder how authors select baselines. Can you discuss the related work ( Multi-Objective Meta Learning, Neurips 2021, and Reasonable Effectiveness of Random Weighting: A Litmus Test for Multi-Task Learning, TMLR). The first work involves comparing the weighting methods under DMTL or MTAN.

4. Lacking details on GTA4 datasets.

**Limitations:**

See Questions

**Strengths And Weaknesses:**

Strengths:

The idea is clear, and the writing is easy to follow.

The result is impressive. A clear performance improvement.

Weaknesses:

Additional ablation study is needed.

Need more baselines

---

> ### Author Rebuttal · Authors · 2026-03-29
>
> We are truly encouraged by your positive assessment of our work across all four dimensions (soundness, presentation, significance, and originality), and we appreciate the opportunity to further strengthen the empirical support.
>
> **W1/Q1: Ablations on the loss functions**
>
> To verify the contribution of each objective, we conducted a component-wise ablation on NYUv2, removing one major component at a time while keeping the rest of the training setup unchanged.
>
> |Method|mIoU↑|Pix Acc↑|Abs Err↓|Rel Err↓|Mean↓|Med↓|11.25°↑|22.5°↑|30°↑|
> |-|-|-|-|-|-|-|-|-|-|
> |w/o Reconstruction|0.4734|0.7235|0.4365|0.1837|28.11|23.49|0.2499|0.4822|0.6086|
> |w/o CFA|0.5547|0.7630|0.3876|0.1636|23.91|18.52|0.3165|0.5814|0.7041|
> |w/o CKA|0.5636|0.7719|0.3555|**0.1433**|22.98|17.22|0.3416|0.6090|0.7255|
> |**Ours**|**0.5693**|**0.7759**|**0.3544**|0.1466|**22.49**|**16.83**|**0.3501**|**0.6196**|**0.7360**|
>
> **Q2: Discrepancy between (Rel Err) and (Abs Err) on Cityscapes**
>
> We thank the reviewer for raising this important question about the gap between Rel Err and Abs Err. We carefully re-checked the evaluation code and further verified the result with a depth-bin analysis. This discrepancy comes from the depth distribution of Cityscapes and the different behaviors of the two metrics. As shown below, **48.55% of valid pixels** fall into the low-depth interval [0, 0.10).
>
> Hence, in the low-depth regime, very similar absolute errors can lead to very different relative errors because Rel Err = $|y - \hat{y}|/ y$. In [0, 0.10), our Abs Err is nearly identical to STCH (0.0114 vs. 0.0116), but the Rel Err is much lower (40.32 vs. 88.16). Since this interval accounts for nearly half of all valid pixels, it strongly affects the final Rel Err while having much less impact on the overall Abs Err.
>
> This pattern is also consistent with the role of CORE-MTL. By separating geometry-related semantics ($Z_s$) from residual appearance factors ($Z_r$), the prediction stream is less affected by nuisance appearance variation and can better preserve fine-grained geometric structure in the low-depth regime. As a result, the overall Abs Err remains close to strong baselines, whereas the overall Rel Err improves substantially.
>
> |Depth Bin|Pixels %|Ours Abs Err|STCH Abs Err|Ours Rel Err|STCH Rel Err|
> |-|-|-|-|-|-|
> |[0.00, 0.10)|48.55%|0.0114|0.0116|**40.3151**|88.1640|
> |[0.10, 0.25)|30.91%|0.0140|0.0137|0.0901|0.0884|
> |[0.25, 0.50)|20.54%|0.0118|0.0117|0.0389|0.0364|
> |**OVERALL**|**100.00%**|**0.0123**|**0.0123**|**19.6088**|**42.8389**|
>
> **W2/Q3: Discussion of related work (MTAN, STCH, RLW, MOML)**
>
> We thank the reviewer for this important suggestion. We agree that the original presentation did not make the baseline selection logic sufficiently clear, especially because **MTAN** and **STCH** belong to different categories. Our comparison set covers the major paradigms in modern multi-task learning: **architecture-centric methods** that modulate feature sharing (e.g., **MTAN**); **optimization-centric methods**, including **weighting-based or balancing methods** (e.g., **RLW**, **GradNorm**), **gradient surgery or manipulation methods** (e.g., **PCGrad**, **FairGrad**), **scalarization-based multi-objective optimization methods** (e.g., **STCH**), and **meta-optimization methods** (e.g., **MOML**); and recent **representation-level methods** (e.g., **RepMTL**).
>
> We will revise the related-work discussion accordingly. In the experiments, we additionally report **MOML**, **RLW**, **FairGrad**, and **RepMTL** on **NYUv2** here as a representative benchmark. Due to the rebuttal space limit, we do not include the full tables for the other benchmarks, but the corresponding comparisons have **also been completed** and will be included in the revised manuscript.
>
> |Method|mIoU|PixAcc|AbsErr|RelErr|Mean|Med|11.25|22.5|30|
> |-|-|-|-|-|-|-|-|-|-|
> |MOML (NeurIPS 2021)|0.5303|0.7496|0.3952|0.1612|24.0679|17.4130|0.3469|0.5987|0.7093|
> |RLW (TMLR 2022)|0.5293|0.7510|0.3883|0.1601|24.0588|17.2890|0.3465|0.6033|0.7134|
> |ExcessMTL (ICML 2024)|0.4846|0.7180|0.3877|0.1624|26.1446|19.8459|0.3043|0.5499|0.6676|
> |FairGrad (ICML 2024)|0.5291|0.7427|0.3944|0.1585|23.0717|**16.3569**|**0.3657**|**0.6239**|0.7312|
> |RepMTL (ICCV 2025)|0.5492|0.7598|0.3727|0.1503|24.5348|19.6139|0.3018|0.5576|0.6847|
> |**Ours**|**0.5693**|**0.7759**|**0.3544**|**0.1466**|**22.4927**|16.8337|0.3501|0.6196|**0.7360**|
>
> **Q4: GTA5 dataset and Sim-to-Real setup**
>
> The GTA5→Cityscapes experiment is a standard sim-to-real transfer setting under the LibMTL protocol, where GTA5 is used as the synthetic source domain and Cityscapes as the real target domain. We use 24,966 synthetic source images and map the semantic labels to the 7-class system to match our Cityscapes setup. For depth, we follow the uncalibrated inverse-depth protocol to account for the scale discrepancy between domains. We will clarify these dataset and evaluation details in the revised manuscript and Appendix F.

---

> > ### Author Rebuttal · Reviewer_mVbL · 2026-04-02
> >
> > Thank you for your reply, I will keep my score.

---

> > > ### Author Response · Authors · 2026-04-07
> > >
> > > We sincerely thank you for your positive evaluation of our work and for your thoughtful feedback during the rebuttal process. We greatly appreciate your recognition and are committed to carefully incorporating your constructive suggestions into the final version of the paper. We would be happy to provide any further clarifications or additional details if needed. We truly value your support and look forward to addressing any remaining questions to ensure all concerns are fully resolved.

---

### Official Review · Reviewer_U9FS · 2026-03-13

**Soundness:** 3
**Presentation:** 2
**Significance:** 3
**Originality:** 3
**Overall Recommendation:** 3
**Confidence:** 3

**Summary:**

This paper proposes CORE-MTL to address the structural entanglement problem in multi-task learning, where the shared representation mixes semantic information with nuisance/style information. The method separates the shared representation into a semantic stream and a residual stream, so that task-relevant information is encoded in the semantic stream, while nuisance variations such as background, lighting, and texture are captured in the residual stream. It uses a Dual-Stream Encoder and Counterfactual Augmentation to promote this separation, and further encourages disentanglement through a grounding mechanism. The proposed method shows strong accuracy on the NYUv2 and Cityscapes datasets.

**Compliance With Llm Reviewing Policy:**

Affirmed.

**Key Questions For Authors:**

Can the authors provide a more detailed explanation of the Physics-Based Inverse Rendering Decoder?
Can the authors include more recent and broader baselines for comparison?

**Limitations:**

The explanation of the Physics-Based Inverse Rendering Decoder is not sufficiently detailed relative to the importance of the claim it supports. Because this decoder is central to grounding the semantic/residual split, the lack of architectural and mechanistic clarity weakens the paper’s causal interpretation.

Although the paper makes a strong causal argument, the experimental evidence for causal disentanglement itself remains somewhat indirect. The current results support robustness and improved performance, but they do not fully establish that the learned decomposition corresponds to a causally valid separation of semantic and nuisance factors.

The comparison set is somewhat narrow and is still centered mainly on classical optimization-centric baselines. As a result, it is difficult to assess how much of the gain comes from the proposed causal-disentanglement framework relative to more recent representation-level MTL approaches.

**Strengths And Weaknesses:**

Strength 1. The paper presents a clear and interesting shift in perspective by arguing that negative transfer in multi-task learning should be addressed at the representation level, rather than only through gradient balancing or gradient surgery.

Strength 2. The proposed framework is technically structured in a coherent way. The combination of a dual-stream encoder, CKA-based independence regularization, counterfactual augmentation, and grounding via reconstruction forms a unified pipeline for encouraging disentanglement between task-relevant and nuisance information.

Strength 3. The paper reports strong empirical results on both in-distribution and out-of-distribution settings.

Weakness 1. In Figure 1, the notation $\tilde{z_s}$ is not explained either in the figure itself or in the main text.

Weakness 2. The explanation of the Physics-Based Inverse Rendering Decoder is not sufficiently clear. This is a critical issue, because it is difficult to logically verify whether the disentanglement of $Z_s$ and $Z_r$ is actually established in a causal manner.

Weakness 3. Among the recent methods used for comparison, only one was published in 2024. The proposed method should be compared against a broader set of more recent approaches.

---

> ### Author Rebuttal · Authors · 2026-03-28
>
> We thank the reviewer for the careful and technically grounded comments. We especially appreciate the reviewer’s thoughtful distinction between strong empirical performance and the strength of the causal interpretation.
>
> **W1: Clarification of notation ($\tilde{z}_s$).**
>
> In Fig. 1, ($\tilde{z}_s$)  simply denotes the semantic representation re-encoded from the synthesized counterfactual image ($\tilde{x}$). It is shown separately only to make clear that, after changing the residual factor to generate ($\tilde{x}$), we **pass** ($\tilde{x}$) **through the shared encoder again** and apply the task heads to its **semantic stream**, checking whether prediction remains stable under residual intervention. This corresponds to ($[\Phi_θ(\tilde{x})]_s$) in Eq. (12). We will clarify this in Fig. 1 and Sec. 3.1.
>
> **W2/Q1/L1/L2: Role of the Physics-Based Inverse Rendering Decoder ($D_{phy}$).**
>
> We agree that the original manuscript did not explain ($D_{phy}$) clearly enough relative to the importance of the claim it supports. Our intention is **not** to claim that ($D_{phy}$) by itself proves full causal identification. Rather, it provides a **structured grounding mechanism** in our framework, because independence alone may still permit an arbitrary latent split. At a high level, the semantic stream is intended to capture task-relevant, geometry-related structure that remains stable under interventions, while the residual/photometric side captures appearance-related variation such as albedo and illumination. In this sense, the role of ($D_{phy}$) is to make the semantic/residual factorization **less arbitrary** during reconstruction, rather than to serve as a standalone proof that the learned factors exactly match true causal semantic and nuisance variables.
>
> Concretely, we use a grounding decoder with the explicit image-formation rule $\hat{x} \approx A(\hat{z}_r) \odot S(N(\hat{z}_s), L(\hat{z}_r))$.
>
> This imposes an **asymmetric role assignment** during reconstruction: geometry-related structure is explained through the semantic pathway, while photometric variation is handled on the residual side. Thus, $D_{phy}$ does **not** by itself establish a fully causally valid separation, but it reduces identifiability ambiguity and provides **stronger semantic anchoring** than independence or generic reconstruction alone.
>
> Architecturally, $D_{phy}$ is **not** a generic decoder over concatenated latents. It uses **separate prediction heads** for distinct physical factors: normals are predicted from the semantic stream, while the photometric side is realized through two complementary pathways—a spatial photometric latent for albedo and a highly compressed global illumination code for lighting. These quantities are combined through a **fixed, non-learned** physics-based shading module (second-order spherical harmonics with Lambertian composition), rather than a generic neural renderer. Compared with a generic decoder, this imposes a more structured geometry-versus-photometry decomposition during reconstruction. This interpretation is also consistent with our added hard-vs.-generic grounding comparison: the full model with $D_{phy}$ is stronger overall, although we do **not** treat this as standalone proof of full causal validity.
>
> **Table A: In-Distribution (ID) NYUv2 Results**
> |Method|mIoU↑|PixAcc↑|AbsErr↓|RelErr↓|Mean↓|Med↓|11.25°↑|22.5°↑|30°↑|
> |-|-|-|-|-|-|-|-|-|-|
> |w/o $D_{phy}$|0.5652|0.7715|0.3555|**0.1427**|23.0044|17.3100|0.3415|0.6068|0.7238|
> |**Full**|**0.5693**|**0.7759**|**0.3544**|0.1466|**22.4927**|**16.8337**|**0.3501**|**0.6196**|**0.7360**|
> We will revise the paper to make both this high-level role and the corresponding architecture explicit, and to calibrate the causal wording more carefully.
>
> **W3/Q2/L3: Extension of baselines and representation-level comparison**
>
> We expanded the comparison set beyond the original optimization-centric baselines by adding **five** additional methods spanning **2021--2025**: **MOML (NeurIPS 2021), RLW (TMLR 2022), FairGrad (ICML 2024), ExcessMTL (ICML 2024), and RepMTL (ICCV 2025)**. Notably, **RepMTL (ICCV 2025)** is a recent **representation-level** MTL baseline, directly addressing this concern.
>
> |Method |mIoU↑ |PixAcc↑ |AbsErr↓ |RelErr↓ |Mean↓ |Med↓ |11.25°↑ | 22.5°↑ | 30°↑ |
> |-|-|-|-|-|-|-|-|- |-|
> |MOML (NeurIPS 2021)|0.5303|0.7496|0.3952|0.1612|24.0679|17.4130|0.3469|0.5987|0.7093|
> |RLW (TMLR 2022) | 0.5293 | 0.7510 | 0.3883 | 0.1601 | 24.0588 | 17.2890 | 0.3465 | 0.6033 | 0.7134 |
> |ExcessMTL (ICML 2024) | 0.4846 | 0.7180 | 0.3877 | 0.1624 | 26.1446 | 19.8459 | 0.3043 | 0.5499 | 0.6676 |
> |FairGrad (ICML 2024) | 0.5291 | 0.7427 | 0.3944 | 0.1585 | 23.0717 | 16.3569 |  **0.3657** |  **0.6239** | 0.7312 |
> |RepMTL (ICCV 2025) | 0.5492 | 0.7598 | 0.3727 | 0.1503 | 24.5348 | 19.6139 | 0.3018 | 0.5576 | 0.6847 |
> | **Ours** | **0.5693** | **0.7759** | **0.3544** | **0.1466** | **22.4927** | **16.8337** |0.3501 | 0.6196 | **0.7360** |

---

> > ### Author Rebuttal · Reviewer_U9FS · 2026-04-04
> >
> > The rebuttal provides a substantially clearer explanation of the intended role of the Physics-Based Inverse Rendering Decoder and appropriately tempers the original causal claim. In particular, it is helpful that the authors clarify that the decoder is not intended to establish full causal identification by itself, but rather to serve as a structured grounding mechanism that reduces identifiability ambiguity and encourages a less arbitrary semantic/residual split. That said, the evidence remains somewhat indirect, and the mechanism is still not specified in enough detail to fully verify that the decomposition is established in a genuinely causal manner. Thus, while the rebuttal meaningfully addresses the concern and improves the interpretability of the proposed decoder, it does not fully resolve the underlying uncertainty regarding the strength of the causal interpretation. For this reason, my overall assessment remains unchanged.

---

> > > ### Author Response · Authors · 2026-04-05
> > >
> > > We thank the reviewer for the careful follow-up. The more precise reading of our paper is **causally motivated**, rather than a claim that the learned decomposition is established in a **strictly causal** or fully identifiable manner. Our causal content is therefore **conditional on an assumed latent Structural Causal Model (SCM)** and remains at the **representation level**: under this view, semantic–nuisance entanglement is a structural source of negative transfer, and reducing such entanglement improves OOD robustness and mitigates task interference beyond what post-hoc gradient surgery alone can guarantee. This calibration is important because, as prior work has emphasized, **strict recovery of true causal latent factors is generally not available from raw observational data alone** [1,2]. Accordingly, our method is **not** claimed to recover the true causal variables from pixels.
> > >
> > > Under this narrower interpretation, the role of the Physics-Based Inverse Rendering Decoder should be understood more precisely. It is **neither** a proof of full causal identification **nor** a blind latent regularizer. Rather, it acts as **structured grounding / factor-role anchoring under the assumed latent SCM**. The key distinction is that **a blind regularizer acts globally**, whereas our decoder imposes a **directional, role-specific constraint** through the image-formation prior: it encourages the semantic stream to support stable geometry/task-relevant structure, while the residual stream explains complementary photometric/style variation needed for reconstruction and counterfactual recombination (Fig. 2). In this sense, the learned decomposition should be interpreted as a **less-arbitrary, structurally grounded semantic–residual split**, rather than as proof of a strictly causal separation of the true underlying factors (Table 3 / Fig. 3). In other words, the contribution of ($D_{\text{phy}}$) is to make this split more structurally anchored rather than merely increasing regularization strength.
> > >
> > > This narrower interpretation also clarifies how our evidence should be read. The current results support robustness, improved performance, and evidence for a more structurally anchored decomposition, but they should not be read as establishing strict causal validity in the identification sense. **To avoid a stronger-than-intended reading, we will revise the manuscript accordingly so that the presentation consistently reflects this narrower interpretation.**
> > >
> > > [1] Locatello et al., Challenging Common Assumptions in the Unsupervised Learning of Disentangled Representations, ICML 2019.
> > >
> > > [2] Brehmer et al., Weakly Supervised Causal Representation Learning, NeurIPS 2022.

---

### Decision · Program_Chairs · 2026-04-30

**Decision:**

Accept (regular)

**Comment:**

This paper proposes CORE-MTL for dense multi-task learning, aiming to reduce negative transfer by disentangling task-relevant semantic information from nuisance variation in the shared representation.
Reviewers generally agreed that the idea is reasonable and that the results on NYUv2 and Cityscapes are strong. During rebuttal, the authors clarified the role of the reconstruction decoder and softened the causal claims, which addressed several of the concerns. As a result, while one reviewer remained unconvinced and maintained a negative recommendation, the others were generally more supportive and leaned toward acceptance.
Overall, I recommend weak acceptance, with the expectation that the revised paper should make the rebuttal clarifications more explicit in the final presentation.